

# Land potential assessment and trend-analysis using 2000–2021 FAPAR monthly time-series at 250 m spatial resolution

Julia Hackländer[1,2], Leandro Parente[1], Yu-Feng Ho[1], Tomislav Hengl[1], Rolf Simoes[1], Davide Consoli[1], Murat Şahin[1], Xuemeng Tian[1,2], Martin Jung[3], Martin Herold[2,4], Gregory Duveiller[5], Melanie Weynants[5] and Ichsani Wheeler[1]

[1] OpenGeoHub, Wageningen, Netherlands
[2] Wageningen University and Research, Wageningen, Netherlands
[3] Biodiversity, Ecology and Conservation Research Group, International Institute for Applied Systems Analysis (IIASA), Laxenburg, Austria
[4] Helmholtz GFZ German Research Centre for Geosciences, Remote Sensing and Geoinformatics, Potsdam, Germany
[5] Max Planck Institute for Biogeochemistry (MPI-BGC), Jena, Germany

Corresponding author
Julia Hackländer,
julia.hacklaender@opengeohub.org

## ABSTRACT

The article presents results of using remote sensing images and machine learning to map and assess land potential based on time-series of potential Fraction of Absorbed Photosynthetically Active Radiation (FAPAR) composites. Land potential here refers to the potential vegetation productivity in the hypothetical absence of short–term anthropogenic influence, such as intensive agriculture and urbanization. Knowledge on this ecological land potential could support the assessment of levels of land degradation as well as restoration potentials. Monthly aggregated FAPAR time-series of three percentiles (0.05, 0.50 and 0.95 probability) at 250 m spatial resolution were derived from the 8-day GLASS FAPAR V6 product for 2000–2021 and used to determine long-term trends in FAPAR, as well as to model potential FAPAR in the absence of human pressure. CCa 3 million training points sampled from 12,500 locations across the globe were overlaid with 68 bio-physical variables representing climate, terrain, landform, and vegetation cover, as well as several variables representing human pressure including: population count, cropland intensity, nightlights and a human footprint index. The training points were used in an ensemble machine learning model that stacks three base learners (extremely randomized trees, gradient descended trees and artificial neural network) using a linear regressor as meta-learner. The potential FAPAR was then projected by removing the impact of urbanization and intensive agriculture in the covariate layers. The results of strict cross-validation show that the global distribution of FAPAR can be explained with an $R^2$ of 0.89, with the most important covariates being growing season length, forest cover indicator and annual precipitation. From this model, a global map of potential monthly FAPAR for the recent year (2021) was produced, and used to predict gaps in actual *vs.* potential FAPAR. The produced global maps of actual *vs.* potential FAPAR and long-term trends were each spatially matched with stable and transitional land cover classes. The assessment showed large negative

![PeerJ](PeerJ logo)

FAPAR gaps (actual lower than potential) for classes: urban, needle-leave deciduous trees, and flooded shrub or herbaceous cover, while strong negative FAPAR trends were found for classes: urban, sparse vegetation and rainfed cropland. On the other hand, classes: irrigated or post-flooded cropland, tree cover mixed leaf type, and broad-leave deciduous showed largely positive trends. The framework allows land managers to assess potential land degradation from two aspects: as an actual declining trend in observed FAPAR and as a difference between actual and potential vegetation FAPAR.

## INTRODUCTION

Land productivity has been identified by the United Nations Convention to Combat Desertification (UNCCD) as one of the key indicators in land degradation assessments (*Sims et al., 2021*). It resembles the productive biological capacity of the terrestrial ecosystem and provides a measure of vegetation health and ecosystem functioning. Identifying degradation processes as a result of different human management options requires continuous monitoring of land productivity, resulting in time- and resource-intensive costs, especially when validating involved processes on the ground. Remote sensing has therefore emerged as a more efficient alternative to long-term controlled field trials, with the ability to allow objective wall-to-wall mapping and continuous monitoring of changes over time.

In recent decades, researchers have focused on modeling global land productivity and land degradation using time series of earth observation (EO) biophysical indices such as the Normalized Difference Vegetation Index (NDVI), Leaf Area Index (LAI), and similar (*De Jong et al., 2011*; *Gonzalez-Roglich et al., 2019*; *Rotllan-Puig, Ivits & Cherlet, 2021*; *Teich et al., 2019*; *Venter et al., 2020*). *Sims et al. (2021)*, building on the work of *Ivits & Cherlet (2013)*, proposed using three indicators to assess land productivity based on a 15–year baseline period and an analysis of: (i) significant trend, (ii) change in current state compared to the past state, and (iii) difference in state compared to ecologically similar units in the surrounding area. For (iii), *Gonzalez-Roglich et al. (2019)* used unique intersections of land cover and soil type to derive ecologically similar units, while *Rotllan-Puig, Ivits & Cherlet (2021)* used a *k*-means clustering combined with a local net scaling approach based on phenological variables (*Prince, Becker-Reshef & Rishmawi, 2009*). In principle, all the approaches listed above use some form of spatial stratification to estimate land productivity, assuming that a comparison of land productivity with past conditions or with ecologically similar units should reveal its level of degradation. However, a baseline period of *e.g.*, 15 years may be insufficient to capture the full ecological potential of the land as realized in deep time (*Enquist et al., 2020*). In addition, the separation into ecologically similar units is largely based on informed, yet subjective

categorization. Furthermore, the surrounding environment used for the reference comparison may already be in a degraded state, leaving the assessor prone to the effect commonly referred to as "shifting baseline syndrome" (*Papworth et al., 2009*). Alternatively, comparisons with an assumed natural or "maximum ecological state", *i.e.*, a deep–time state in the absence of short–term anthropogenic influence, could reveal a land's full ecological potential (*Enquist et al., 2020*).

The concept of "potential natural vegetation" (PNV) has first been defined by *Tüxen (1956)* as the vegetation that would be encountered in the absence of human intervention and of any significant changes to the current climatic conditions. The PNV concept has previously been applied in various studies by modeling the potential distribution of natural biomes (*Bonannella et al., 2023*; *Hengl et al., 2018*; *Levavasseur et al., 2012*), and by mapping the potential biomass stocks of forests (*Erb et al., 2018*; *Roebroek et al., 2023*). *Hengl et al. (2018)*, specifically, proposed a data-driven framework using a machine learning approach to estimate potential primary productivity. Previous studies on potential primary productivity, such as from *Hengl et al. (2018)* and *Krause et al. (2022)*, however, are limited by the coarse spatial resolution of EO data, and/or by focusing on a single region or a short time period. In this article, we apply the PNV approach to derive a new baseline reference for land monitoring and to provide estimates of the potential for land productivity. We follow a data-driven approach based on spatio-temporal machine learning to assess the global primary productivity of vegetation and analyze trends. For this, we make use of time series data of the Fraction of Absorbed Photosynthetically Active Radiation (FAPAR). FAPAR is defined as the fraction of the incoming solar radiation in the spectral range of 400–700 nm that is absorbed by vegetation, hence, values depend to a large extent on canopy and leaf properties, and can in turn be associated with vegetation biomass (*Mõttus et al., 2012*). It is a unitless parameter, that can take values between 0 and 1. FAPAR provides a direct, bio-physically-based metric of land productivity. As a result, FAPAR plays a key role in primary productivity models (*Fuster et al., 2020*; *Verger et al., 2023*; *Zhao et al., 2005*), and has been recognized as both an Essential Climate Variable (*Zemp et al., 2022*); as well as an Essential Biodiversity Indicator (*Skidmore et al., 2021*). *Hengl et al. (2018)* simulated the potential FAPAR for the time period 2014–2017 using observed "natural" vegetation sampled from protected areas as the reference state. However, protected areas are subject to varying definitions, commonly allow anthropogenic land uses within them, and are difficult to validate on the ground as ecologically intact. *Shao et al. (2024)* showed that variables related to human activity, such as land use and population density, can support the identification of degradation degree. Hence, removing the impact of such variables could give estimates of the PNV state of the land. Here, we train the FAPAR model on globally available biophysical and human pressure variables, without selecting or removing points that do not fall in natural vegetation areas. Next, we apply the model to project the hypothetical state of no-human-impact by imposing the least human pressure under the same biophysical conditions. FAPAR under no-human pressure in our study refers to the potential FAPAR we could observe when removing the impact of urbanization, intensive agriculture, fertilization, and irrigation. Knowledge of potential primary productivity can then be used to fundamentally

underpin agricultural diversification and substantiate land restoration targets; as well as to monitor restoration efforts at the field level.

Several global FAPAR products derived from remote sensing technology exist, such as products of Moderate Resolution Imaging Spectroradiometer (MODIS) (*Knyazikhin et al., 1998*), Global LAnd Surface Satellite (GLASS) (*Liang et al., 2021*), GEOV2 (*Verger et al., 2023*), Medium Resolution Imaging Spectrometer (MERIS) (*Bacour et al., 2006*), Visible Infrared Radiometer Suite (VIIRS) (*Yan et al., 2018*), PRoject for On-Board Autonomy–Vegetation (PROBA-V) (*Fuster et al., 2020*), and Ocean and Land Colour Instrument (OLCI) (*Gobron et al., 2022*) among others. *Ma et al. (2022)* have recently produced a consistent and harmonized time series of FAPAR at 250 m resolution covering 2000 to 2021 in 8–day interval steps by integrating several satellite products: the existing MODIS Collection 6, PROBA-V V1, and GLASS V5 FAPAR products. This product showed improvements compared to GLASS V5 FAPAR in terms of higher spatio-temporal continuity and details in reflectance values. In this article, we make use of this new state–of–the–art global FAPAR dataset to derive trends and identify differences in current and potential land productivity, in order to demonstrate its potential for land degradation assessments and use in land restoration efforts. Our main research questions include:

**RQ1** What are the key environmental variables explaining spatial and temporal variation in monthly FAPAR?

**RQ2** What is the maximum achievable accuracy of predicting the monthly FAPAR?

**RQ3** Which land cover types show the largest gaps between actual *vs* potential FAPAR?

**RQ4** Which land cover types show positive and negative long–term trends of FAPAR (2000–2021)?

The data and code reported in this work are available under an open data license from https://doi.org/10.5281/zenodo.8381409 and https://doi.org/10.5281/zenodo.8403714; code used to produce analysis and visualizations is available at https://github.com/Open-Earth-Monitor/Global_FAPAR_250m.

## MATERIALS AND METHODS

### General workflow

To model land potential we use the newly provided GLASS FAPAR V6 dataset as a baseline input (*Ma, 2022*). We estimated trends and gaps in FAPAR globally per pixel for the time period 2000–2021 using a spatio-temporal machine learning (ML) framework. An overview of the workflow is shown in Fig. 1. The whole processes can be represented with seven (7) main steps. First, we aggregated the GLASS FAPAR V6 8-day time series from 2000 to 2021 to monthly values of three percentiles (5th, 50th and 95th), representing their position in the value distribution. Second, we generate a sampling design and then overlay the points using the location of the points and the date to produce a regression matrix to model the actual FAPAR as a function of environmental predictors, including several variables related to human pressure. Third, we optimize and fit an ensemble machine learning model. Fourth, we use the fitted model to predict potential FAPAR, in

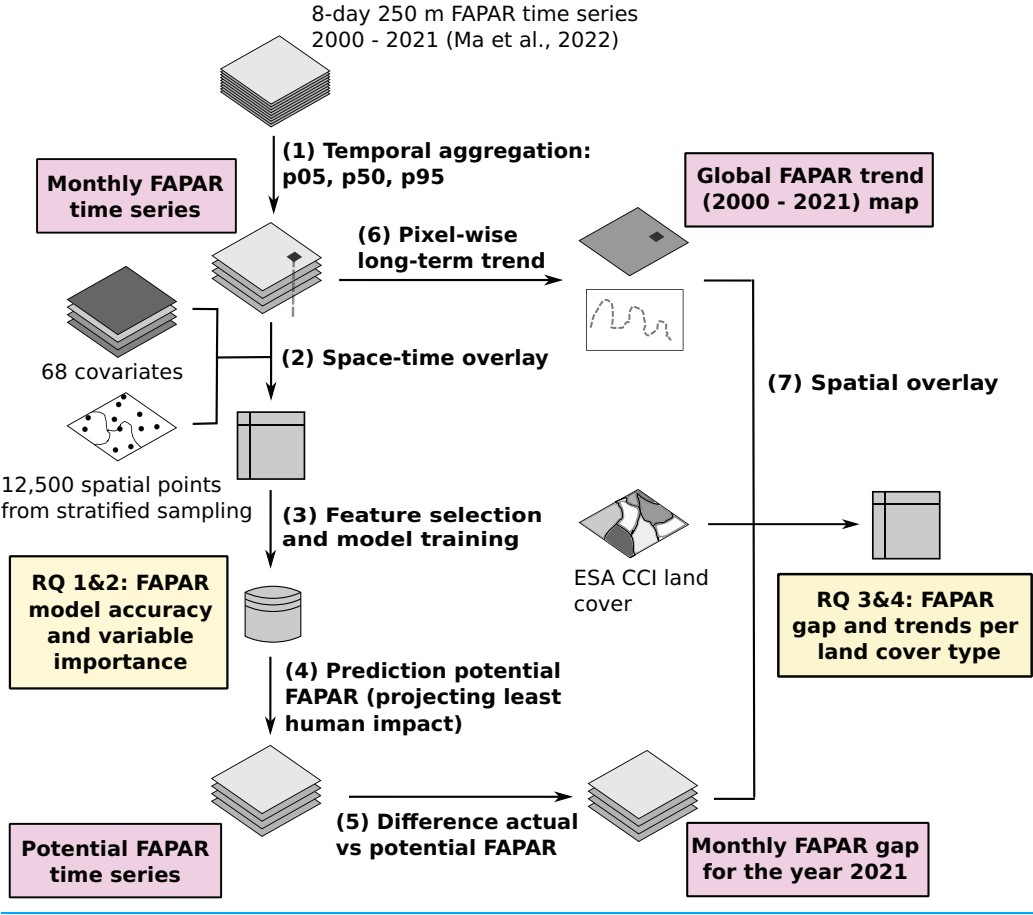

**Figure 1 General workflow of the data analysis and modeling with main inputs, outputs and the main research questions (*Ma et al., 2022*).**

this case by setting the value in the hypothetical state of no-human-impact, *i.e.*, removing the impact of urbanization, intensive fertilization, and irrigation. Fifth, we calculated the gap between actual and potential FAPAR globally per pixel on a monthly basis for the year 2021. Sixth, we derived the long-term linear trend per pixel to supplement the gap analysis between potential and actual FAPAR, and finally (seventh) we ran a spatial overlay and match both the FAPAR long–term trend and the FAPAR gap with land cover maps to derive statistics per land cover type.

## GLASS FAPAR V6

The GLASS FAPAR V6 data is a harmonized and consistent time series of FAPAR at 250 m resolution, which covers the time period from 26th February 2000 to 27th December 2021 in 8 day interval steps (*Ma et al., 2022*). It is an improved version of the GLASS FAPAR V5 in terms of time series trend stability. In contrast to V5, which makes use of radiative transfer functions, V6 is derived from a bidirectional long/short-term memory (Bi–LSTM) model to estimate FAPAR using Moderate Resolution Imaging Spectroradiometer (MODIS) reflectance and LAI data and several other FAPAR products (MODIS Collection 6, GLASS FAPAR V5, and PROBA-V1 FAPAR).

We derived monthly FAPAR time series by extracting, per pixel, the p0.05, p0.50, p0.95 percentiles using time series of the 8–day GLASS FAPAR V6 data. The percentiles indicate the FAPAR values that encompass the specified percentage of data set values falling below them. The 50th percentile (p0.50), hence, represents the median of the aggregated values. We processed all data in MODIS Sinusoidal Tile Grid and using full parallelization as implemented in the `joblib` library version 1.1.0 (https://joblib.readthedocs.io) and the `scikit-map` library (https://github.com/scikit-map/scikit-map) in Python. For each month, we aggregated all composites within that month plus one composite each before and after that month, resulting in five to six composites as input for each monthly value. This helped to reduce potential missing dates in the time series and noise in the original FAPAR 8–day composites and allowed a better understanding of differences within each month. Finally, we merged the processed tiles into global Cloud-Optimized GeoTiffs, which resulted in a total of 792 composites throughout 22 years. These data (almost 1.4 TB in size) are publicly available *via* https://doi.org/10.5281/zenodo.8381409.

We explore the monthly aggregated FAPAR time series by spatially overlaying all training points used for modeling with monthly aggregated time series data of the Enhanced Vegetation Index (EVI) derived from MOD13Q1 (50th percentile) at 250 m spatial (*Didan, 2021*), monthly daytime land surface temperature derived from MOD11A2 (50th percentile) at 1 km spatial resolution (*Wan, Hook & Hulley, 2015*), monthly water vapor derived from MCD19A2 (mean value) at 1 km spatial resolution (*Lyapustin & Wang, 2022*; *Parente, Simoes & Hengl, 2023*), and monthly accumulated precipitation from CHELSA V2.1 (*Karger et al., 2017*, *2021*) at 1 km spatial resolution. To aid interpretation of FAPAR patterns, we created density scatter plots to visualize FAPAR (all three percentiles p0.05, p0.50, p0.95) against each variable and calculated Pearson's r coefficient.

We validated our monthly aggregated GLASS FAPAR V6 values against reference measurements provided by Ground Based Observation for Validation (GBOV). The "RM6 —Fraction of Intercepted Photosynthetically Active Radiation" data was downloaded from https://land.copernicus.eu/global/gbov/ (*Brown et al., 2020*). The stations included in this analysis are shown in Table 1. For sites where measurements from both upward and downward facing DHP (Digital Hemispherical Photography) images were available (FIPAR–up and FIPAR–down, respectively), we approximated total FIPAR following Eq. (1), as provided in *Brown et al. (2020)*:

$$FIPAR = FIPAR_{up} + (1 - FIPAR_{up}) * FIPAR_{down}. \tag{1}$$

Note that the difference between DHP-derived FIPAR and FAPAR is generally considered marginal (*Brown et al., 2020*).

We filtered out any values with a measurement error greater than 0.1 or any raised quality issue (quality flag greater than 1). Each site has differing time intervals between measurements. For a more coherent comparison with the monthly GLASS FAPAR V6 values, we aggregated the reference measurement values per site to monthly averages whenever more than one value per month and year was available. This resulted in 3,262 unique space-time points. Reference measurements, as provided by GBOV, represent one elementary sampling unit each. Depending on the ground station, the extent of an

**Table 1 Ground sites and time periods used for FAPAR reference measurements in this study.** Note that most sites consist of 2–3 individual ground stations in proximity to each other, resulting in a total of 74 individual ground stations.

| Site name | Network | IGBP class | Start date | End date |
|---|---|---|---|---|
| Bartlett experimental forest | NEON | Mixed forest | 2014-05 | 2021-10 |
| Blandy experimental farm | NEON | Deciduous broadleaf | 2015-04 | 2021-10 |
| Central plains experimental range | NEON | Grasslands | 2014-03 | 2021-10 |
| Dead lake | NEON | Deciduous broadleaf | 2016-03 | 2021-10 |
| Guanica forest | NEON | Evergreen broadleaf | 2015-07 | 2021-12 |
| Hainich | FluxNet | Mixed forest | 2019-08 | 2020-10 |
| Harvard forest | NEON | Mixed forest | 2014-05 | 2021-10 |
| Jones ecological research center | NEON | Evergreen needleleaf | 2013-06 | 2016-07 |
| Jornada | NEON | Open shrublands | 2015-06 | 2021-11 |
| Konza prairie biological station | NEON | Croplands | 2017-07 | 2021-10 |
| Lajas experimental station | NEON | Grasslands | 2016-04 | 2021-12 |
| Litchfield savanna | TERN | Woody savannas | 2020-11 | 2022-02 |
| Moab | NEON | Open shrublands | 2015-05 | 2021-09 |
| Niwot ridge mountain research station | NEON | Evergreen needleleaf | 2017-07 | 2021-09 |
| Oak ridge | NEON | Mixed forest | 2015-03 | 2021-10 |
| Onaqui ault | NEON | Open shrublands | 2014-05 | 2021-09 |
| Ordway swisher biological station | NEON | Evergreen needleleaf | 2013-04 | 2021-12 |
| Santa Rita | NEON | Closed shrublands | 2016-04 | 2021-10 |
| Smithsonian conservation biology institute | NEON | Mixed forest | 2014-06 | 2021-11 |
| Smithsonian environmental research center | NEON | Croplands | 2015-04 | 2021-10 |
| Steigerwaldt land services | NEON | Deciduous broadleaf | 2015-07 | 2021-09 |
| Talladega national forest | NEON | Evergreen needleleaf | 2014-05 | 2021-10 |
| Tumbarumba | FluxNet | Evergreen broadleaf | 2019-07 | 2020-12 |
| Underc | NEON | Mixed forest | 2015-09 | 2021-10 |
| Valencia anchor station | SM | Cropland mosaics | 2019-07 | 2021-02 |
| Woodworth | NEON | Grasslands | 2014-05 | 2015-10 |

elementary sampling unit can range from 20 m × 20 m to 40 m × 40 m (see the Algorithm Theoretical Basis Document on RM6 provided by the GBOV service at https://land.copernicus.eu/global/gbov/products/). We spatially and temporally matched the stations with GLASS FAPAR V6 values, assuming a relatively homogeneous vegetation cover in the 250 m radius around each station and hence a neglectable effect of differing spatial resolutions between reference and observed data.

## Predictor variables

A total of 68 spatially explicit and harmonized variables representing (bio)climatic, topographic, geographic, vegetation cover and human pressure factors were used for modeling purposes. All layers were resampled onto a standard grid covering latitudes between 87.37°N and 62.0°S and reprojected to the coordinate reference system
EPSG: 4326. The majority of covariates, for example, variables from the Digital Terrain Model (DTM) variables, lights at night, snow probability and similar, were available at 250–500 m spatial resolution (*Yamazaki et al., 2017*). Other 1–km resolution variables were resampled to 250 m resolution using a cubic splines in GDAL (*Warmerdam, 2008*). Whenever time-series covariate data were unavailable for the years 2020 and/or 2021, we used the most recent available year (*e.g.*, previous year). We also added the percentile value of FAPAR as a covariate so that the fitted model can be used to predict FAPAR for an arbitrary part of the distribution.

### Topographic variables

Elevation and derived variables originate from a global MERIT DEM at 250 m spatial resolution (*Yamazaki et al., 2019*). From this dataset we included derived variables: elevation, slope, sine and cosine of aspect, upslope and downslope curvature, negative and positive openness, compound topographic index and valley bottom flatness.

### Climatic variables

All climate data was acquired or derived from CHELSA V2.1 at 1 km spatial resolution (*Brun et al., 2022a*, *2022b*; *Karger et al., 2017*, *2021*) that was downscaled to 250 m spatial resolution using cubic splines using GDAL. Specifically, we made use of:

- 15 BIOCLIM variables and two climatologies (growing season length (gsl) and snow cover days (scd)) for the time period from 1981–2010. BIOCLIM variables BIO8, BIO9, BIO18, BIO19 and other climatologies were excluded either due to artifacts or spatial gaps, *i.e.*, being incomplete.
- Time series of monthly average, minimum, and maximum air temperature, long-term monthly values, and the difference of actual and long-term values for each date.
- Time series of monthly precipitation sum, long-term monthly values, and the difference of actual and long-term values for each date.
- Time series of cumulative annual precipitation.

### Lithologic and landform variables

Lithological and landform classification variables based on the USGS Global Ecosystem Map (*Hartmann & Moosdorf, 2012*) were downloaded as indicator maps at 250 m spatial resolution (*Hengl, 2018*). Lithological indicators (0–100% fraction maps) include: "basic plutonics", "siliciclastic sedimentary", "intermediate plutonics", "acid volcanic", "metamorphics", "intermediate volcanics", "unconsolidated sediment", "carbonate sedimentary rock", "acid plutonics", "pyroclastics", "mixed sedimentary rock", "evaporite", "basic volcanics". Landform indicators (0–100% fraction maps) include: "breaks foothills", "flat plains", "high mountains deep canyons", "hills", "low hills", "low mountains", and "smooth plains".

### Geometric temperature variables

The geometric minimum and maximum temperature is a variable obtained by geometric transformation of latitude, day–of–the–year and elevation (*Kilibarda et al., 2014*).

We include these to represent the geometry of the earth and the day–of–the–year. Geometric temperatures may improve distinct predictions of points showing similar monthly or long-term temperatures in climate time series despite their distinct latitudinal location or seasonal time position. *Witjes*'s *et al. (2022)* work, for example, showed a high feature importance of geometric temperatures in machine learning predictions of land cover classes. The geometric minimum and maximum temperatures, respectively denoted as $t_{min}$ and $t_{max}$, can be defined anywhere on the globe using:

$$t_{min}(day, \phi, z) = 24.2 \cdot \cos \phi - 15.7 \cdot (1 - \cos \theta(day, \phi)) \cdot \sin |\phi| - 0.6 \cdot \frac{z}{100}, \qquad (2)$$

$$t_{max}(day, \phi, z) = 37 \cdot \cos \phi - 15.4 \cdot (1 - \cos \theta(day, \phi)) \cdot \sin |\phi| - 0.6 \cdot \frac{z}{100} \qquad (3)$$

with $\theta$ derived as

$$\theta(day, \phi) = (day - 18) \cdot \frac{2\pi}{365} + 2^{1-\text{sgn}(\phi)} \cdot \pi, \qquad (4)$$

where $day$ is the day of year, $\phi$ is the latitude, $z$ is the elevation in meters, the number 18 represents the coldest day in the northern and warmest day in the southern hemisphere, 0.6 is the vertical temperature gradient per 100 m elevation increase, and sgn denotes the signum function that extracts the sign of a real number.

### Vegetation cover variables

We derived annual indicator maps (values 0 or 100 for absence/presence) from the ESA CCI land cover at 300 m spatial resolution for each vegetation cover type of forest, shrubland, cropland and grassland. Vegetation cover types were aggregated from ESA CCI land cover classes to IPCC classes as described in *ESA (2017)*. The aggregation scheme can be seen in Table 2.

### Human pressure variables

We prepared a number of global layers at moderate resolution (250 m) representing human pressure:

- Yearly Human Footprint Index: The index combines data on land cover, accessibility, nightlights and population density in a scoring scheme to produce annual estimates of human pressure at 1 km spatial resolution (*Mu et al., 2022*).
- Population count: We downloaded 5–year interval population count data at 100 m spatial resolution from *Schiavina et al. (2023)*. We linearly interpolated between each 5–year interval to derive annual population count estimates.
- Cropland intensity: Annual fraction of cropland within each $1 \times 1$ km cell derived from the History Database of the Global Environment 3.2 (HYDE 3.2) and the Land-Use Harmonization 2 (LUH2) (*Cao et al., 2021a, 2021b*).
- Nightlights: Annual visible nightlight V2 from NASA/NOAA Visible Infrared Imaging Radiometer Suite for the time period 2012–2019 (*Elvidge et al., 2021*) and extrapolated for 2000–2011 using logistic regression by *Hengl (2023)* at 500 m spatial resolution.

**Table 2  Crosswalk table from ESA CCI land cover and Biome 6000 classification to vegetation cover variables.**

| Vegetation cover variable | ESA CCI class (ESA, 2017) | ESA CCI code | Biome 6000 class (Hengl et al., 2018) |
|---|---|---|---|
| Forest | Tree cover, broadleaved, evergreen, closed to open (>15%) | 50 | Cold deciduous forest<br>Cold evergreen needleleaf forest |
| | Tree cover, broadleaved, deciduous, closed to open (>15%) | 60, 61, 62 | Cool evergreen needleleaf forest<br>Cool mixed forest |
| | Tree cover, needleleaved, evergreen, closed to open (>15%) | 70, 71, 72 | Cool temperate rainforest<br>Temperate deciduous broadleaf forest |
| | Tree cover, needleleaved, deciduous, closed to open (>15%) | 80, 81, 82 | Temperate evergreen needleleaf open woodland<br>Temperate sclerophyll woodland and shrubland |
| | Tree cover, mixed leaf type (broadleaved and needleleaved) | 90 | Tropical deciduous broadleaf forest and woodland<br>Tropical evergreen broadleaf forest |
| | Mosaic tree and shrub (>50%)/herbaceous cover (<50%) | 100 | Tropical semi evergreen broadleaf forest<br>Warm temperate evergreen and mixed forest |
| | Tree cover, flooded, fresh or brakish water | 160 | |
| | Tree cover, flooded, saline water | 170 | |
| Shrubland | Shrubland | 120 | Xerophytic woods scrub |
| Grassland | Mosaic herbaceous cover (>50%)/tree and shrub (<50%) | 110 | Tropical savanna |
| | Grassland | 130 | – |
| Cropland | Rainfed cropland | 10, 11, 12 | |
| | Irrigated cropland | 20 | |
| | Mosaic cropland (>50%)/natural vegetation (tree, shrub, herbaceous cover) (<50%) | 30 | |

## Training points

We generated training points using a stratified random sampling approach, excluding areas of permanent water or ice (Brus, 2022). To ensure the inclusion of a minimum number of samples from proportionally underrepresented feature-space areas of human impact and of areas with least human impact, we created a set of non-spatially overlapping strata focusing on:

1. areas with minimal human pressure: all four human pressure variables at zero in years at the start (2000 and 2001), and at the end of the time series (2020 and 2021),

2. areas with recent change in vegetation cover from forest/wetland to cropland/grassland and *vice versa* comparing years 2000/2001 to 2019/2020,

3. areas with a change in nightlight values above plus or below minus $1[\text{nW.cm}^{-2}.\text{sr}^{-1}]$ comparing the average of the years 2000/2001 to 2020/2021.

We calculated the optimal sample size per stratum based on the total standard deviation of each stratum of four FAPAR images across the quarters (January, April, July and

**Table 3 Overview of training point strata encoding, sample sizes and sample weights.**

| Stratum code | Nightlights change | Land cover change | Human pressure | Population size | Standard deviation | Sample size | Sample weight |
|---|---|---|---|---|---|---|---|
| 0 | outside land mask (permanent water/ice) | | | – | – | – | – |
| 1 | >/< | 0 | 100 | 23,517,852,472 | 72.02 | 7,738 | 3,039,268 |
| 2 | >/< | 1 | 0 | 13,870,136 | 70.29 | 204 | 67,991 |
| 3 | >/< | 1 | 100 | 174,111,464 | 60.17 | 247 | 704,905 |
| 4 | >/< | 2 | 0 | 11,797,016 | 58.52 | 203 | 58,113 |
| 5 | >/< | 2 | 100 | 163,219,120 | 59.30 | 243 | 671,684 |
| 7 | >1 | 0 | 100 | 529,097,288 | 53.98 | 327 | 1,618,035 |
| 9 | >1 | 1 | 100 | 3,092,984 | 50.65 | 201 | 15,388 |
| 11 | >1 | 2 | 100 | 2,689,992 | 48.41 | 201 | 13,383 |
| 13 | <−1 | 0 | 100 | 270,521,584 | 55.27 | 267 | 1,013,189 |
| 15 | <−1 | 1 | 100 | 1,914,936 | 52.32 | 200 | 9,575 |
| 17 | <−1 | 2 | 100 | 1,278,024 | 49.04 | 200 | 6,390 |
| 18 | any other combination | | | 6,670,463,000 | 76.44 | 2,469 | 2,701,686 |

**Notes:**

Nightlights: >/<, values in the range −1 to 1 (nW.cm$^{-2}$.sr$^{-1}$); >1, values greater than 1; <1, values less than −1.
Land cover code: 0, any other LC type or change than 1 or 2; 1, forest/wetland to grassland/cropland; 2, grassland/cropland to forest/wetland.
Human pressure: 0, all human pressure variables equal to zero; 100, any one of the human pressure variables >0.

October) of the years 2001 and 2021, using Neyman's allocation formula (*Neyman, 1934*; *Wright, 2014*):

$$n_h = \frac{N_h S_h}{\sum_{i=1}^{H} N_i S_i} n,$$ (5)

and calculating sample weights $W_h$ using

$$W_h = \frac{N_h}{n_h},$$ (6)

where $n_h$ is the sample size for stratum $h$, $N_h$ is the population size, $S_h$ is the standard deviation and $n$ is the total sample size.

In summary, a total of 12,500 unique spatial points were sampled using this sampling design. A minimum sample size of 200 was applied to each stratum, and the remaining samples were distributed proportionally according to the standard deviation of each stratum. A full overview of the strata and sample sizes can be found in Table 3. The stratification of points with simple random sampling within each stratum resulted in globally distributed samples (Fig. 2). Each training point was overlaid with the time series of each of the three FAPAR percentiles (p0.05, p0.50, p0.95) and of all covariate layers from 2000–2021. This resulted in a total of 9,825,000 unique space-time data points. In this first model test, the cross-validation of the ANN model exceeded the volatile memory available in the machine used for the computation (*i.e.*, 1 TB), hence we finally decided to randomly subset the 10 millon points to about 3 million space-time points from the total as input for modeling. Note that the regression matrices are also large because we used a large number of covariate layers.

## Potential FAPAR model building and evaluation

### Feature selection

To reduce the risk of overfitting FAPAR models, we ran a recursive feature elimination (RFE) using a random forest estimator (number of trees = 60) and five–fold spatial cross validation (RFECV). Spatial blocking at the training points ensured that the time series data points belonging to the same geographic location are only present in either the training or validation data set. We refer to this as the "strict spatial cross-validation".

To reduce computation costs, we selected a random subset of 4,000 spatially unique locations (3,132,000 space-time points) from the total training point data set. After running RFECV, we ran RFE to obtain the features of the highest importance in predicting FAPAR. From these two outputs, we selected the predictor variables for fitting the actual model.

### FAPAR model training and evaluation

We trained an ensemble machine learning (EML) model (*Brownlee, 2021*) to predict FAPAR as a function of topography, bioclimate, lithology, vegetation cover, and human pressure. The EML model uses a linear regressor to stack three base learners: Extremely randomized trees (Extra Trees), Gradient descended trees (XGBoost), and an artificial neural network (ANN). The EML model was implemented in Python, using the `scikit-map` library built on `scikit-learn` (*Pedregosa et al., 2011*). Due to the limitation of computing resources, we randomly sampled a total of 3,132,000 space-time points from across all 12,500 spatially unique points to ensure a global spatial coverage for training, resulting in a different random subset of the total training points than for recursive feature elimination. We selected the parameters of each model by hyperparameter tuning a predefined set of parameters to test using `HalvingRandomSearchCV` as implemented in *Pedregosa et al. (2011)* (Table 4) with five-fold spatial CV. For hyperparameter tuning, we used a maximum resource value of 20,000. The EML model and each sub-model accuracy was evaluated using all 3,132,000 points in a five-fold spatial CV with spatial blockage on the geographic location. The coefficient of determination ($R^2$), the concordance correlation coefficient (CCC), the root mean squared error (RMSE), mean absolute error (MAE) and bias were calculated using stratum-specific sample weights (Eq.(6)), *i.e.*, the inverse of the inclusion probability, to account for the sampling intensities differing over the strata.

Furthermore, we sampled 500,000 spacetime points from 13,700 independent locations (*i.e.*, locations not included in the RFE, hypertuning, or model training) to evaluate the performance of the final EML model that is trained on all 3,132,000 spacetime points. To keep the validation points representative of the model, independent points were sampled with the same proportions of each stratum as used for the training of the actual FAPAR EML model. The EML model predictions of the actual FAPAR were also validated against the GBOV ground reference measurements as we did for the monthly aggregated GLASS FAPAR V6.

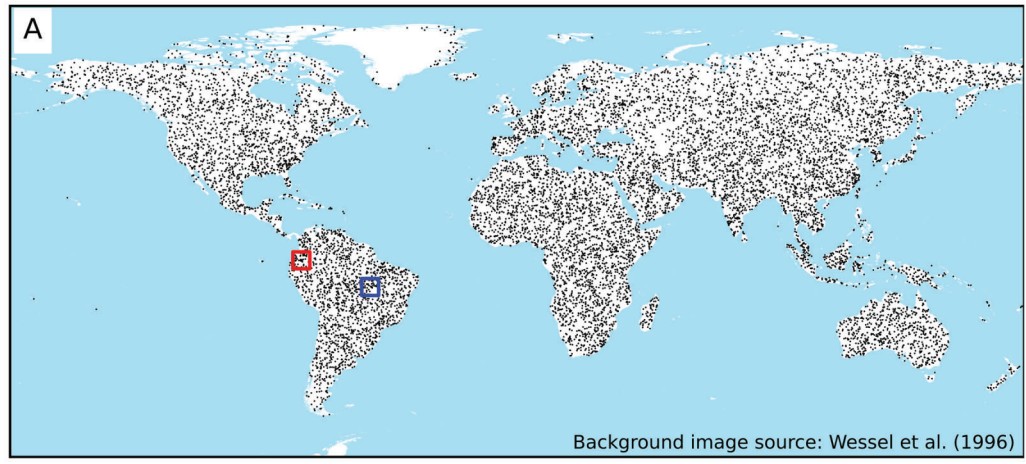

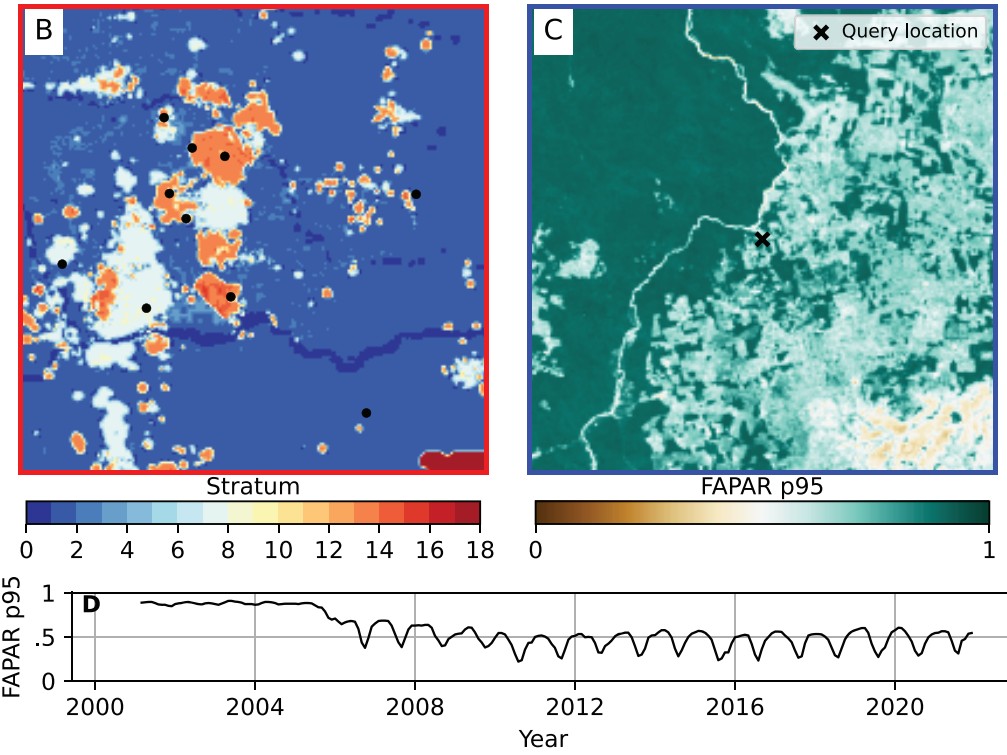

**Figure 2 Global distribution of training points used to model FAPAR.** (A) Global coverage of training points. (B) Example of strata from which the training points were sampled (centered at location lat = 0.24229823, lon = −76.57932387). Source of background image: *Wessel & Smith (1996)*. (C) Example of FAPAR p0.95 at 250 m spatial resolution for December 2021 showing an area of about 300 × 300 km, with a selected point (query location). (D) Time series plot of FAPAR p0.95 monthly values for the query location of (C) at coordinates lat = −9.73096951, lon = −52.15942255, (Brazil) showing a typical deforestation pattern.

## Prediction of potential FAPAR

To model potential FAPAR, we used the FAPAR model trained using all data, but then for prediction, we set a hypothetical space where human impact is set to 0 values, *e.g.*, lights at night, human footprint, cropland intensity, population count. For hypothetical vegetation

**Table 4 Hyperparameter optimisation overview.** Tested and selected hyperparameters for each of the sub-models included in the FAPAR EML meta-estimator.

| Model | Hyperparameter | Lower limit | Upper limit | Selected |
|---|---|---|---|---|
| Extremely randomized trees | Number of estimators | 10 | 100 | 44 |
| | Maximum tree depth | 5 | 100 | 92 |
| | Maximum number of features | 0 | 1 | 0.84 |
| | Minimum samples for splitting | 2 | 100 | 16 |
| | Minimum samples per leaf | 1 | 10 | 2 |
| Gradient descended trees | Number of estimators | 10 | 100 | 81 |
| | Maximum tree depth | 3 | 100 | 50 |
| | Alpha | 0 | 2 | 1.19 |
| | Reg Alpha | 0 | 0.2 | 0.007 |
| | Eta | 0 | 2 | 1.999 |
| | Reg_Lambda | 0 | 0.2 | 0.12 |
| | Gamma | 0 | 2 | 0.05 |
| | Learning rate | 0 | 0.2 | 0.06 |
| | colsample_bytree | 0 | 1 | 0.88 |
| | colsample_bylevel | 0 | 1 | 0.66 |
| | colsample_bynode | 0 | 1 | 0.47 |
| Artificial neural network | Epochs | – | – | 10 |
| | Batch size | – | – | 256 |
| | Learning rate | – | – | 0.0005 |
| | Number of layers | – | – | 4 |
| | Number of neurons | – | – | 128 |
| | Activation | – | – | ReLu |
| | Dropout rate | – | – | 0.15 |
| | Output activation | – | – | Sigmoid |

cover, we use the potential biome distribution of 2000–2017 from *Hengl (2019)*. We use vegetation indicator maps of forest, shrubland, and grassland, and then we translate them into potential vegetation cover maps (Table 2), assigning 0 or 100 for the absence or presence of each biome. At the same time, we set the cropland indicator layer to zero values.

All other covariates were used unmodified in the simulation, thus assuming negligible interactions between vegetation and climatic and topographic variables at the grid level. Spatial predictions were made at 250 m resolution for the 95th percentile. We chose the 95th percentile for predictions of potential FAPAR to reduce the effect of noise in the data. In addition, we provide the standard deviation of the three sub-models, called "model deviance" hereafter, on a per–pixel basis.

## FAPAR gap and trend analysis of time-series data

We calculated the monthly gaps between the actual and the potential FAPAR for the 95th percentile per pixel for the year 2021 by subtracting the potential FAPAR predictions from

the actual FAPAR values. We spatially matched the FAPAR gap with every combination of land cover between 2000 and 2020 according to the ESA CCI annual land cover maps, stable and transition areas (*ESA, 2017*) and visualized the distribution of gap values for each. We conducted the analysis in the aggregated land cover classes as shown in Table 5.

To perform trend analysis, we extracted the trend component of the FAPAR time series (2000–2021) by season–trend decomposition using locally estimated scatterplot smoothing (LOESS), Seasonal and Trend decomposition using Loess (STL) (*Ben Khalfallah et al., 2021*), and fitted an ordinary least squares (OLS) model on the trend component. For every FAPAR pixel we computed the OLS coefficients (alpha and beta), standard deviation, $p$ and $R^2$. The percentage of area with significant positive and negative trends ($p < 0.005$) was calculated for every land cover change combination (2000 and 2020) according to aggregated ESA CCI land cover classes.

# RESULTS

## Monthly aggregated FAPAR data exploration

The correlation between the monthly aggregated actual FAPAR in relation to EVI, precipitation, daytime land surface temperature (LST) and water vapor, together with the calculated Pearson's r for each, is shown in Fig. 3. FAPAR shows a strong positive correlation with EVI (Pearson's $r = 0.89$). The correlation with monthly precipitation and water vapor is also positive but less strong. The density plots indicate a more complex distribution of these two parameters in relation to FAPAR. Water vapor and precipitation both show a similar range of values throughout the FAPAR range. The density plot of LST against FAPAR indicates a decrease of LST with increasing FAPAR. However, this is not reflected in the Pearson's correlation value, which can be attributed to a wide range of values in the LST especially at a FAPAR value of 0, where values range from −50 °C to +50 °C. Generally, FAPAR values in the range 0.75–0.85 were underrepresented compared to the entire FAPAR range. In general, it can be said that FAPAR is strongly correlated with EVI, LST, precipitation, and water vapor, but the relationship is often non-linear and saturation can be observed for FAPAR values near to 0.0 and 0.9.

The validation of the monthly FAPAR shows a strong agreement and a low bias with the GBOV ground validation points ($R^2 = 0.80$, Fig. 4). This matches the results of *Ma et al. (2022)*.

## FAPAR model performance and variable importance

The recursive feature elimination showed an optimal feature number of 52 (out of 68 variables provided). The spatial cross-validation results for modeling distribution of monthly FAPAR with the selected 52 variables using the EML model showed an $R^2$ of 0.89 and a CCC of 0.94. RMSE and MAE resulted in moderate values of 0.10 and 0.06, respectively. The selected hyperparameter sets for the "Extremely randomized trees" and "Gradient descended trees" model are shown in Table 4. The accuracy metrics for each submodel are only slightly lower, showing that each model performs similarly well (Table 6).

**Table 5 Aggregation of land cover classes of ESA CCI for computing summary statistics on FAPAR gap and trend.**

| ESA CCI class (level 4) | Aggregated ESA CCI class |
|---|---|
| 10. Cropland, rainfed | 10. Cropland rainfed |
| 11. Herbaceous cover | 10. Cropland rainfed |
| 12. Tree or shrub cover | 10. Cropland rainfed |
| 20. Cropland, irrigated or post-flooding | 20. Cropland, irrigated or post-flooding |
| 30. Mosaic cropland (>50%), natural vegetation (tree, shrub, herbaceous cover) (<50%) | 30. Mosaic cropland—natural vegetation |
| 40. Mosaic natural vegetation (tree, shrub, herbaceous cover) (>50%), cropland (<50%) | 30. Mosaic cropland—natural vegetation |
| 50. Tree cover, broadleaved, evergreen, closed to open (>15%) | 50. Tree cover broadleaved evergreen |
| 60. Tree cover, broadleaved, deciduous, closed to open (>15%) | 60. Tree cover broadleaved deciduous |
| 61. Tree cover, broadleaved, deciduous, closed (>40%) | 60. Tree cover broadleaved deciduous |
| 62. Tree cover, broadleaved, deciduous, open (15–40%) | 60. Tree cover broadleaved deciduous |
| 70. Tree cover, needleleaved, evergreen, closed to open (>15%) | 70. Tree cover needleleaved evergreen |
| 71. Tree cover, needleleaved, evergreen, closed (>40%) | 70. Tree cover needleleaved evergreen |
| 72. Tree cover, needleleaved, evergreen, open (15–40%) | 70. Tree cover needleleaved evergreen |
| 80. Tree cover, needleleaved, deciduous, closed to open (>15%) | 80. Tree cover needleleaved deciduous |
| 81. Tree cover, needleleaved, deciduous, closed (>40%) | 80. Tree cover needleleaved deciduous |
| 82. Tree cover, needleleaved, deciduous, open (15–40%) | 80. Tree cover needleleaved deciduous |
| 90. Tree cover, mixed leaf type (broadleaved and needleleaved) | 90. Tree cover mixed leaf type |
| 100. Mosaic tree and shrub (>50%) herbaceous cover (<50%) | 100. Mosaic tree and shrub—herbaceous cover |
| 110. Mosaic herbaceous cover (>50%) tree and shrub (<50%) | 100. Mosaic tree and shrub—herbaceous cover |
| 120. Shrubland | 120. Shrubland |
| 121. Evergreen shrubland | 120. Shrubland |
| 122. Deciduous shrubland | 120. Shrubland |
| 130. Grassland | 130. Grassland |
| 140. Lichens and mosses | 140. Lichens and mosses |
| 150. Sparse vegetation (tree, shrub, herbaceous cover) (<15%) | 150. Sparse vegetation |
| 151. Sparse shrub (<15%) | 150. Sparse vegetation |
| 152. Sparse herbaceous cover (<15%) | 150. Sparse vegetation |
| 160. Tree cover, flooded, fresh or brakish water | 160. Tree cover flooded |
| 170. Tree cover, flooded, saline water | 160. Tree cover flooded |
| 180. Shrub or herbaceous cover, flooded, freshsalinebrakish water | 180. Shrub or herbaceous cover flooded |
| 190. Urban areas | 190. Urban areas |
| 200. Bare areas | 200. Bare areas |
| 201. Consolidated bare areas | 200. Bare areas |
| 202. Unconsolidated bare areas | 200. Bare areas |
| 210. Water bodies | 210. Water bodies |
| 220. Permanent snow and ice | 220. Permanent snow and ice |

The bias for all tested models based on five–fold cross validation (CV) is close to zero. The evaluation of the EML model against independent validation set (500,000 spacetime points) resulted in similar values to the spatial CV results with an $R^2$ of 0.90, CCC of 0.95, RMSE of 0.09 and MAE of 0.06 (Fig. 5). The distribution of the density plot of reference
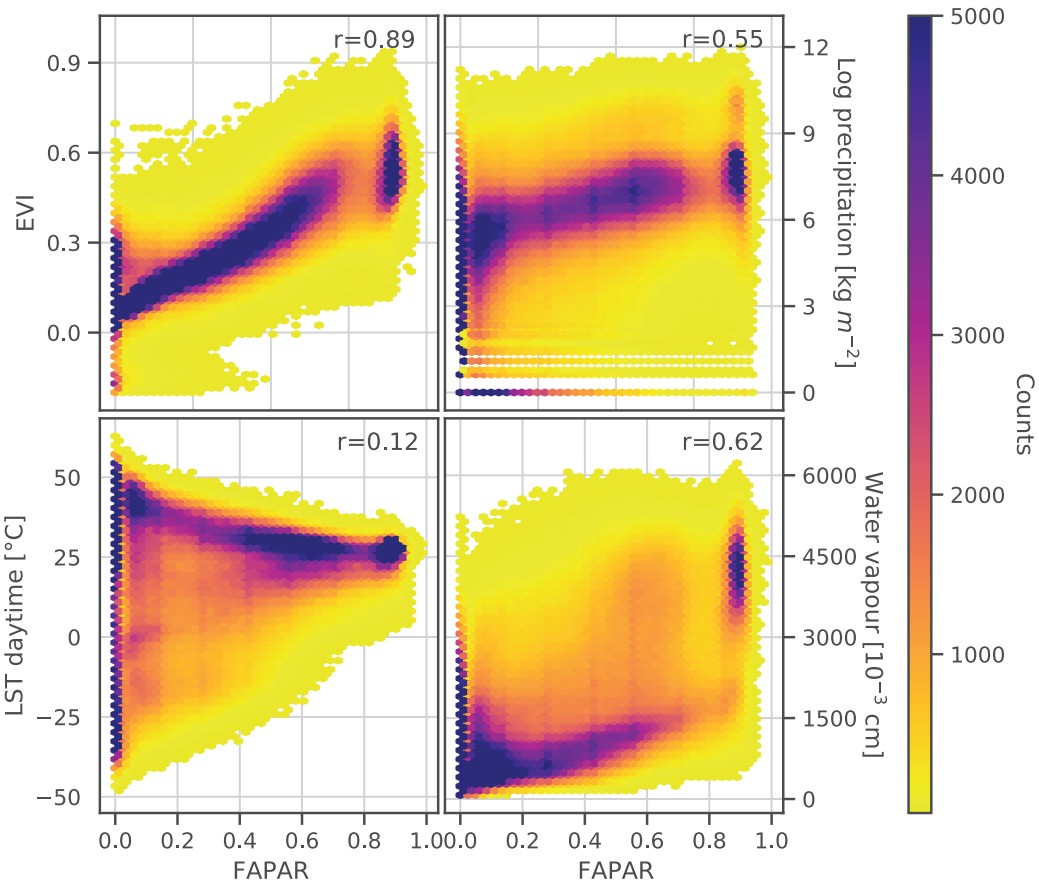

**Figure 3 Density scatter plots and Pearson's r coefficient of monthly GLASS FAPAR V6 p0.50 vs. monthly EVI p0.50, average accumulated precipitation, daytime land surface temperature (LST) p0.50 and average water vapour.**

against prediction for this independent point data set from GLASS FAPAR V6 shows an overall high agreement between the reference and predicted values.

The predicted actual FAPAR at ground stations also showed a high agreement with the GBOV points ($R^2$ = 0.75, Fig. 4). The results of both independent tests show that high FAPAR values (above 0.75) tend to be underestimated, while low FAPAR values (below 0.4) tend to be overestimated (Fig. 5).

Variable importance analysis showed that the three most important variables for modeling were: (1) the growing season length, (2) the forest cover indicator, and (3) the annual precipitation amount (BIO12) as shown in Fig. 6. In general, the climatic variables played a more significant role in explaining FAPAR than topographic and lithologic variables. All variables on human pressure (human footprint index, population count, cropland intensity, and nightlights) were retained in the recursive feature elimination, highlighting the additional information each of them provides in explaining FAPAR, despite appearing lower in the variable importance ranking than climatic variables. While both the "Extremely randomized trees" and the "Gradient descended trees" model showed a higher importance for the human footprint index than other human pressure variables,

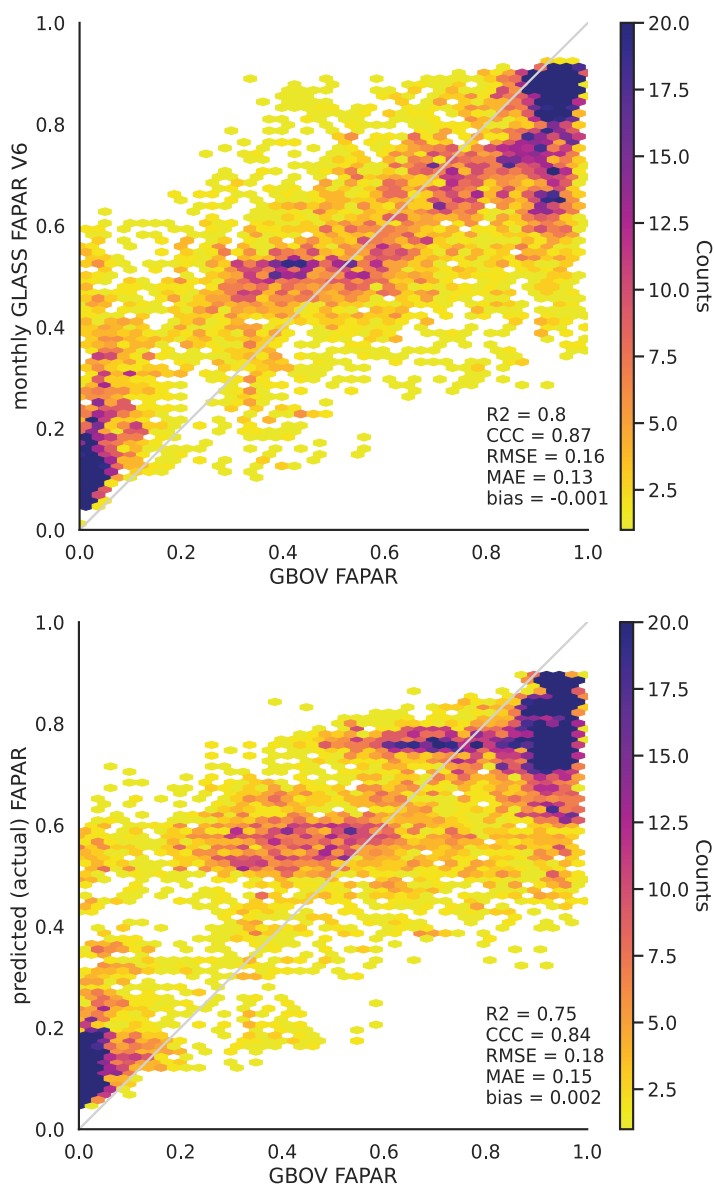

**Figure 4** Monthly aggregated FAPAR (p0.05, p0.50, p0.95) from *Ma et al. (2022)* (upper) and EML predicted actual FAPAR (lower) plotted against monthly average FAPAR derived from GBOV ground measurements.

**Table 6 Evaluation metrics of ensemble model (EML) and each sub-model based on five-fold spatial CV.**

| Model | $R^2$ | CCC | RMSE | MAE | Bias |
|---|---|---|---|---|---|
| EML | 0.89 | 0.94 | 0.10 | 0.06 | −0.0008 |
| Extremely randomized trees | 0.88 | 0.94 | 0.10 | 0.07 | 0.0002 |
| Gradient descended trees | 0.88 | 0.94 | 0.10 | 0.07 | 0.001 |
| Artificial neural network | 0.87 | 0.93 | 0.11 | 0.07 | 0.0001 |

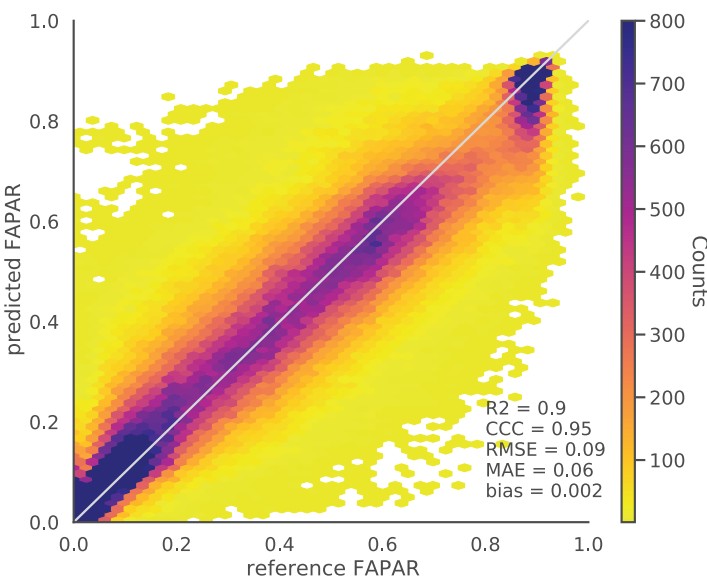

**Figure 5  Density scatter plot of reference *vs*. EML predicted actual FAPAR of independent test data points (independent locations) of GLASS FAPAR V6.**

the "Gradient descended trees" gave cropland intensity, population density and nightlights a higher importance than "Extremely randomized trees".

## Visual examination of global maps of FAPAR gap actual *vs*. potential and long-term trend

We visually inspected the global maps of average annual actual and potential FAPAR produced for the year 2021 (Fig. 7). The global distribution of potential FAPAR indicates that Northern latitudes (Europe, North America, Northern Asia) show mostly negative gap values (actual FAPAR lower than potential), while highest values are found at latitudes close to the equator, especially in the Amazon region, the Congo River Basin and South-East Asia. The model deviance (standard deviation of the sub-models) of these regions is comparably low with other regions of the globe.

The gap of actual *vs*. potential FAPAR shows that the area of these same regions exhibits mostly little difference, but with patches of potential FAPAR lower than actual. Regions with a potential FAPAR lower than the actual relate mainly to locations that show a potential forest cover according to the potential biome maps. On the other hand, regions with a potential FAPAR higher than the actual relate to the distribution of potential grassland and shrub cover.

The model deviance seems to be higher for regions that show a potential FAPAR lower than the actual FAPAR than *vice versa* (Fig. 7). Predictions of potential FAPAR in deforestation areas in Brazil show a potential FAPAR close to the surrounding forested areas, although generally with slightly lower values. This is expected from the model results, which show a tendency to underestimate FAPAR in the range 0.75–1.0 (Fig. 5). We specifically looked at known areas of irrigation to identify the behavior of the model in these areas. The typical pivot irrigation areas in the U.S. and the region around the Nile in

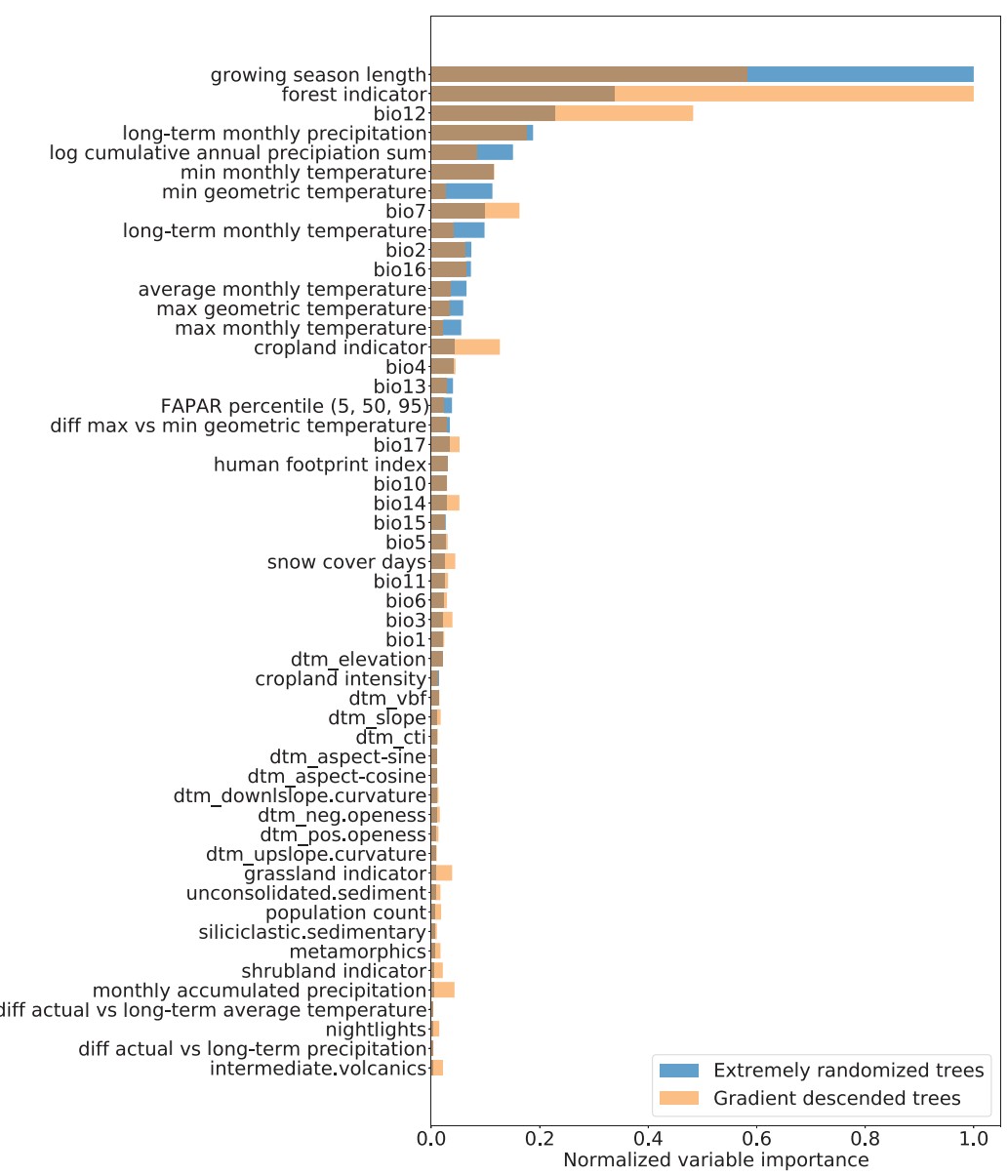

**Figure 6 Normalized variable importance of extremely randomized trees and gradient descended trees models.**

Egypt show a potential FAPAR lower than the actual FAPAR. This shows that the model detects human pressure variables (*e.g.*, cropland intensity) as an important factor in predicting FAPAR.

The global FAPAR trend map for 2000–2021 shows that regions at latitudes close to the equator show large areas of negative trends (Fig. 8). Negative trends are also observed on a wide scale in Australia, North America, and Central Asia. Europe, on the other hand, shows mainly stable and positive trends. Both India and China show especially wide areas of positive FAPAR trends. While in India we observe mostly an actual FAPAR higher than

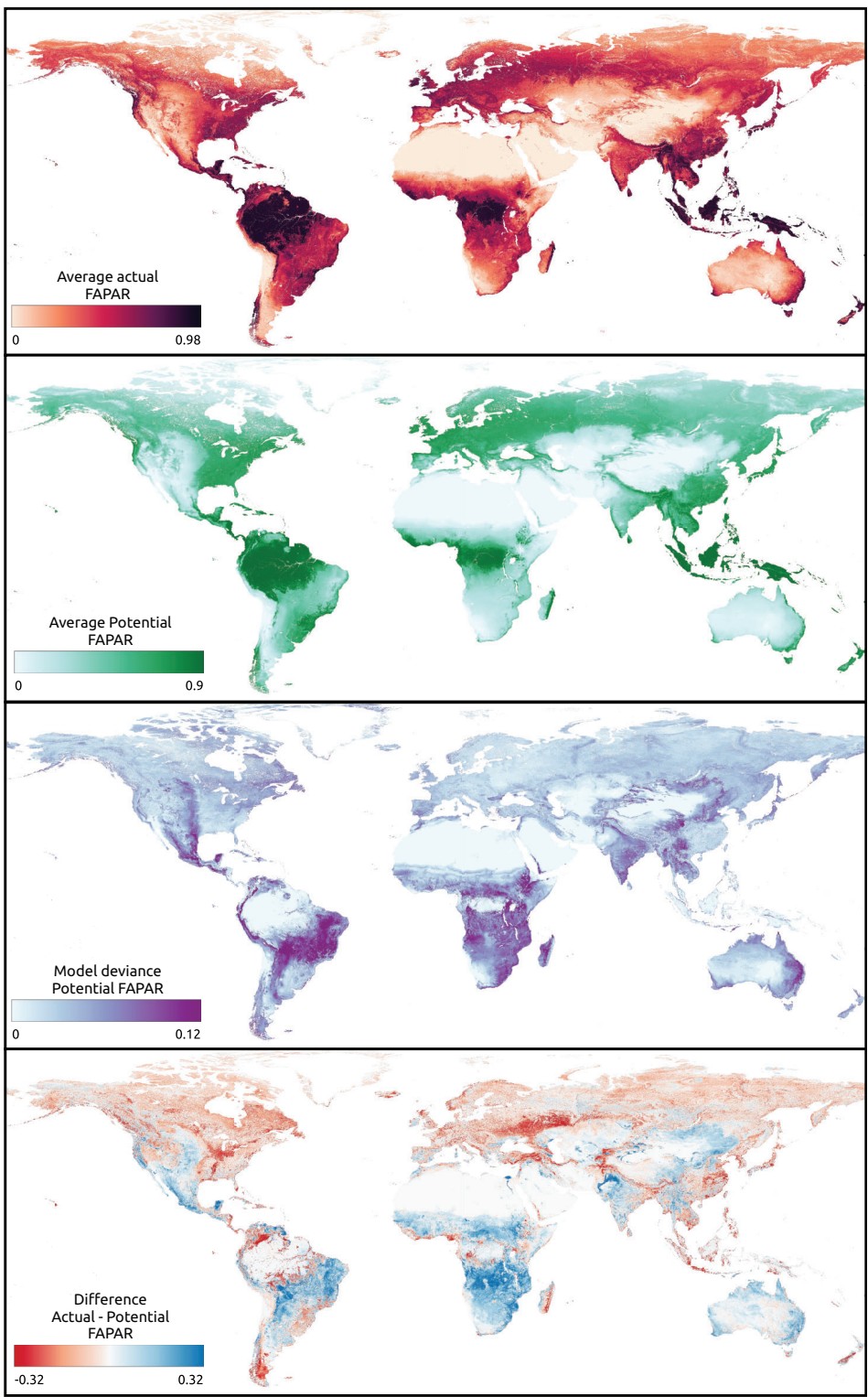

**Figure 7** Global maps of actual, potential, model deviance (standard deviation) and the gap of actual *vs.* potential FAPAR (https://doi.org/10.5281/zenodo.8403714). In the difference map of actual *vs.* potential FAPAR, negative values indicate that actual FAPAR is lower than potential FAPAR.

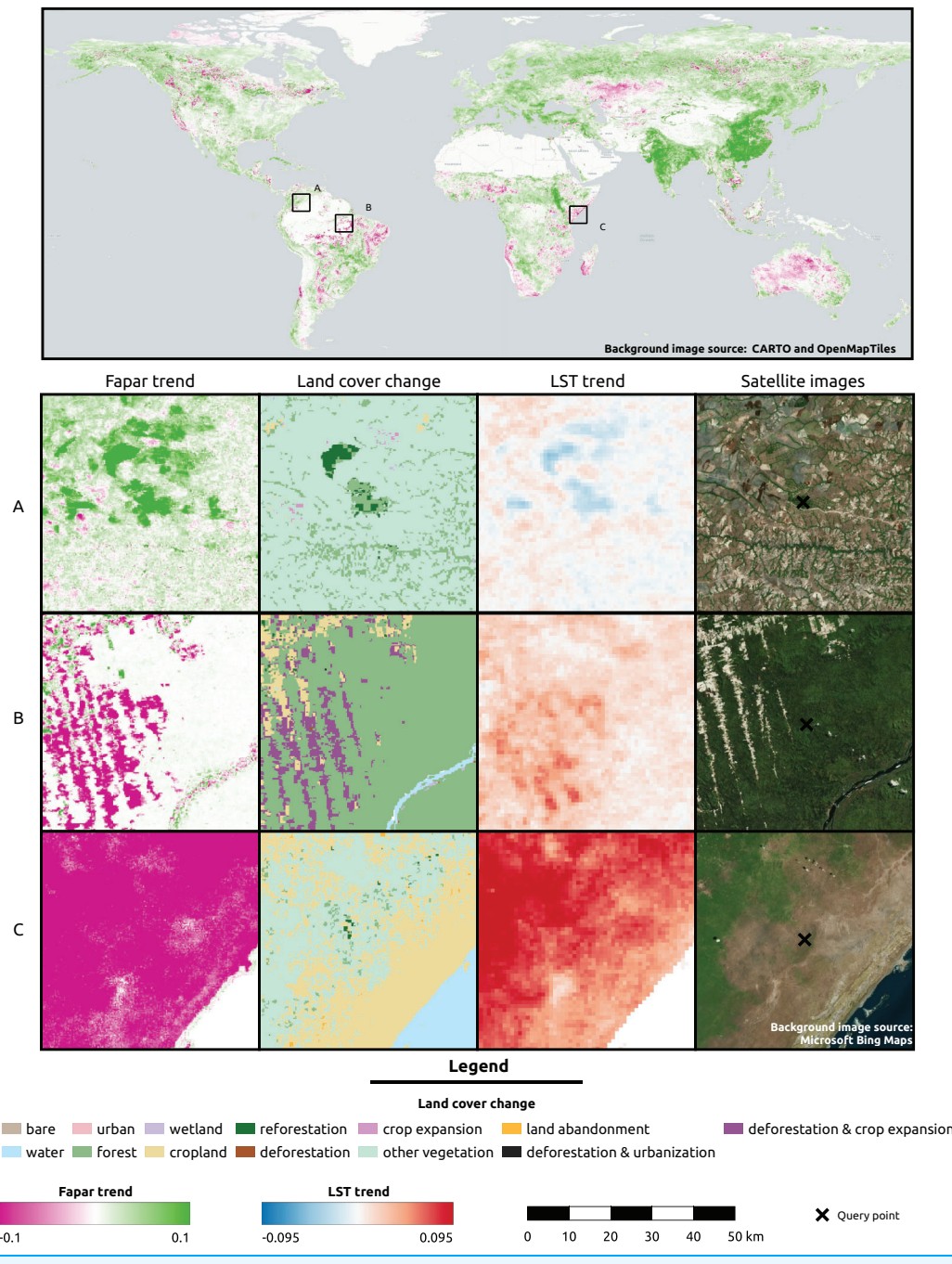

**Figure 8 Global FAPAR trend and zoomed-in examples of three locations showing the spatial overlap with land cover change and LST trend.** (A) Colombia—Las Gaviotas in Cumaribo (lat = 4.53477280, lon = −70.90632447), (B) Brazil—Iriri River in State of Parà (lat = −3.89600839, lon = −53.41974961), (C) Somalia—Kismaayo (lat = −0.39853, lon = 42.26388). Land cover change classes were aggregated from ESA CCI-LC classes according to the CORINE land cover change grid. Gradient legends show cumulative cut count values as min/max. Background image source world map: CARTO (https://carto.com/) and OpenMapTiles.org. Source of satellite image: Microsoft(R) BingTM Maps.

its potential with a high model deviance of the potential FAPAR, in China we observe mostly an actual FAPAR lower than its potential with a low model deviance.

For three sample locations in Figs. 8–10, we provide a more detailed inspection of FAPAR trend, gap and time series. The location of the reforestation project at Las Gaviotas in Colombia shows a spatial overlap of positive FAPAR trend with reforestation and a decrease in LST (Fig. 8). Although the actual FAPAR shows lower values than its potential at the start of the time series, it increases over time to similar values as the potential and even grows beyond the potential values (Fig. 9). The FAPAR maps show that the potential FAPAR is similarly high across the region, while the actual FAPAR is lower in the vicinity of the reforestation area than the potential FAPAR (Fig. 10). The location close to the Iriri River in Brazil shows a spatial overlap of the negative trend of FAPAR with deforestation and an increase in LST. While the potential FAPAR remains stable over time, the actual FAPAR decreases from 2008 to 2016. Similarly, as for the site in Colombia, we can see a homogeneously high potential FAPAR in the area. The gap between actual *vs.* potential FAPAR is small in the stable forest area, while there is large gap in the deforestation part.

For the location of Kismaayo in Somalia, we observe a strong negative trend over the entire inspected area. Land cover remains stable over the period of time; however, an increase in LST observed, which is mostly observed in areas of "other vegetation" (shrubland in this case), while cropland shows a more stable LST over time. The centre pixel shows a decrease in the time series, whereas the potential FAPAR remains rather stable. The actual FAPAR is mostly lower than the potential; however, the model deviance is higher in this area than in the other two sample locations.

## FAPAR gap and trend analysis with land cover

Our spatial analysis of the relationship between the average FAPAR gap for the year 2021 and land cover stable and change classes shows that the largest negative gaps (actual FAPAR lower than potential) are found in the stable classes 'Lichens and mosses' (LICMO) (90%), 'Urban areas' (URBAN) (86%), 'Tree cover needleleaved deciduous' (TREND) (67%), and 'Shrub or herbaceous cover flooded' (SHHFL) (61%); and in the change classes 'Cropland rainfed-Urban areas' (CRPRF-URBAN) (82%), 'Tree cover needleleaved evergreen-Shrubland' (TRENE-SHRUB) (72%), 'Tree cover needleleaved deciduous-Mosaic tree and shrub or herbaceous cover' (TREND-MTSHH) (72%), and 'Tree cover needleleaved evergreen-Mosaic tree and shrub or herbaceous cover' (TRENE-MTSHH) (71%) (see Fig. 11). The largest positive gaps (actual FAPAR higher than potential) are found in the stable classes 'Tree cover broadleaved deciduous' (TREBD) (64%), 'Shrubland' (SHRUB) (60%), and 'Cropland, irrigated or post-flooding' (CRPIF) (57%); and in the change classes 'Tree cover broadleaved deciduous-Shrubland' (TREBD-SHRUB) (87%), 'Shrubland-Tree cover broadleaved deciduous' (SHRUB-TREBD) (84%), 'Shrubland-Cropland rainfed' (SHRUB-CRPRF) (66%), and 'Mosaic cropland or natural vegetation-Tree cover broadleaved deciduous' (MCRNV-TREBD) (53%). Especially classes 'Water bodies' (WATER) and 'Urban areas' (URBAN) show a wider distribution of the gap values, while classes 'Lichens and mosses' (LICMO) and 'Tree cover needleleaved deciduous' (TREND) show a more narrow dispersion.

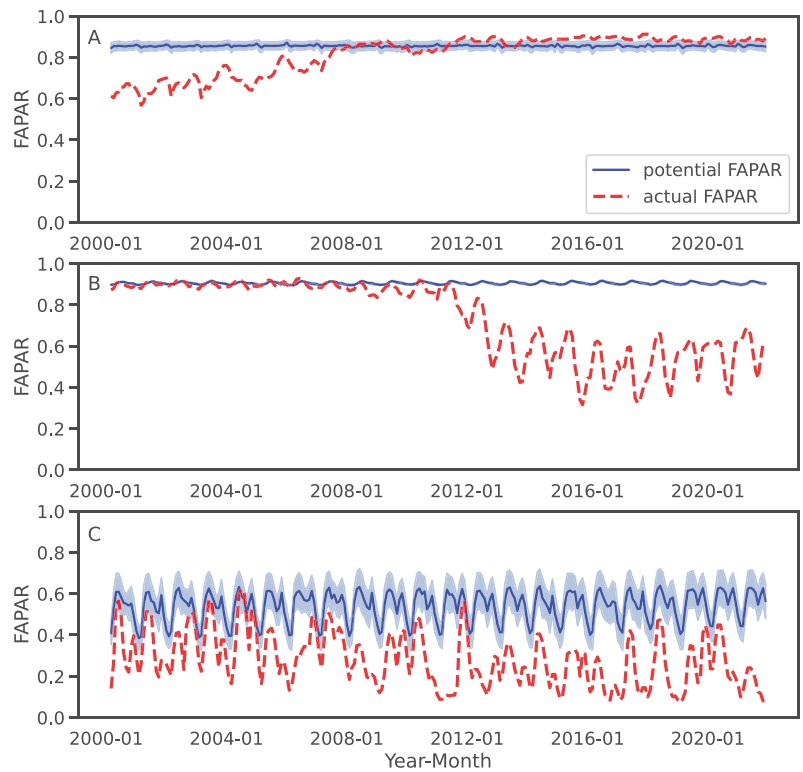

**Figure 9 Examples of FAPAR actual and potential time series 2000–2021 for the three query locations in Fig. 8: The potential FAPAR shows the model deviance (standard deviation) as shaded area.** (A) Colombia (reforestation project Las Gaviotas; lat = 4.53477280, lon = −70.90632447), (B) Brazil (lat = −3.89600839, lon = −53.41974961), (C) Somalia (lat = −0.39853, lon = 42.26388).

Regarding the relation between the FAPAR long–term trend 2000–2021 and land cover change grid, especially high positive values are found for stable classes 'Cropland, irrigated or post-flooding' (CRPIF) and 'Tree cover mixed leaf type' (TREMX); and change classes 'Mosaic cropland or natural vegetation-Tree cover broadleaved evergreen' (MCRNV-TREBE) and 'Mosaic cropland or natural vegetation-Mosaic tree and shrub or herbaceous cover' (MCRNV-MTSHH) (see Fig. 12). Strong negative trends are found for stable classes 'Sparse vegetation' (SPARV) and 'Urban areas' (URBAN); and change classes 'Tree cover broadleaved evergreen-Mosaic cropland or natural vegetation' (TREBE-MCRNV) and 'Tree cover broadleaved evergreen-Cropland rainfed' (TREBE-CRPRF). Land cover classes that show the largest percentage of positive trends within the class are 'Cropland, irrigated or post-flooding' (CRPIF) (85%), 'Tree cover mixed leaf type' (TREMX) (82%), 'Tree cover broadleaved deciduous' (TREBD) (80%), and 'Tree cover needleleaved deciduous' (TREND) (79%). Land cover classes that show the largest percentage of negative trends within the class are 'Sparse vegetation' (SPARV) (33%) and 'Urban areas' (URBAN) (31%). In total, we found that 19% of the global land area shows a decreasing trend, while 67% shows an increasing trend.

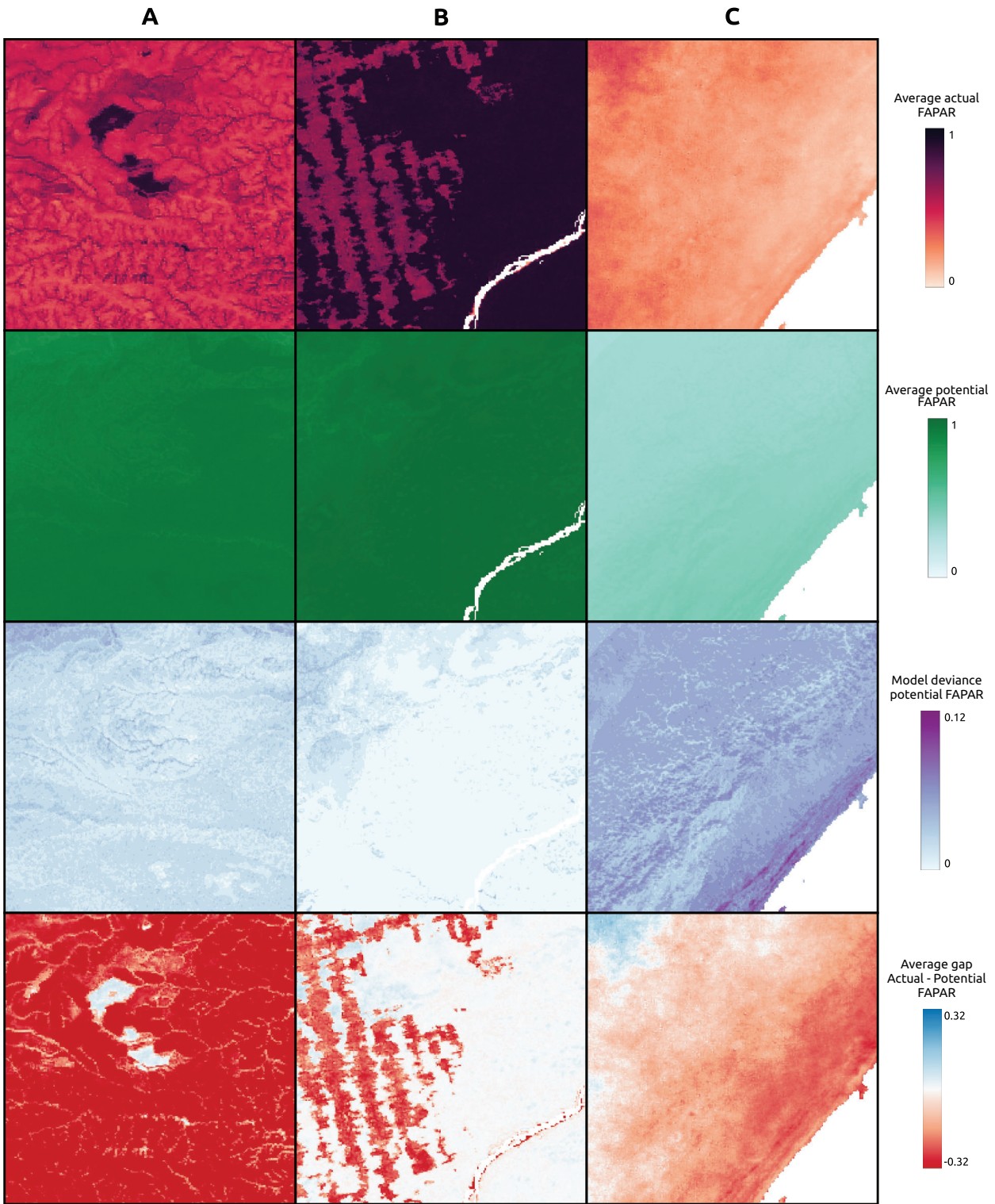

**Figure 10 FAPAR actual, potential and gap examples of example locations.** (A) Colombia–Las Gaviotas in Cumaribo (lat = 4.53477280, lon = −70.90632447), (B) Brazil—Iriri River in State of Parà (lat = −3.89600839, lon = −53.41974961), (C) Somalia—Kismaayo (lat = −0.39853, lon = 42.26388).

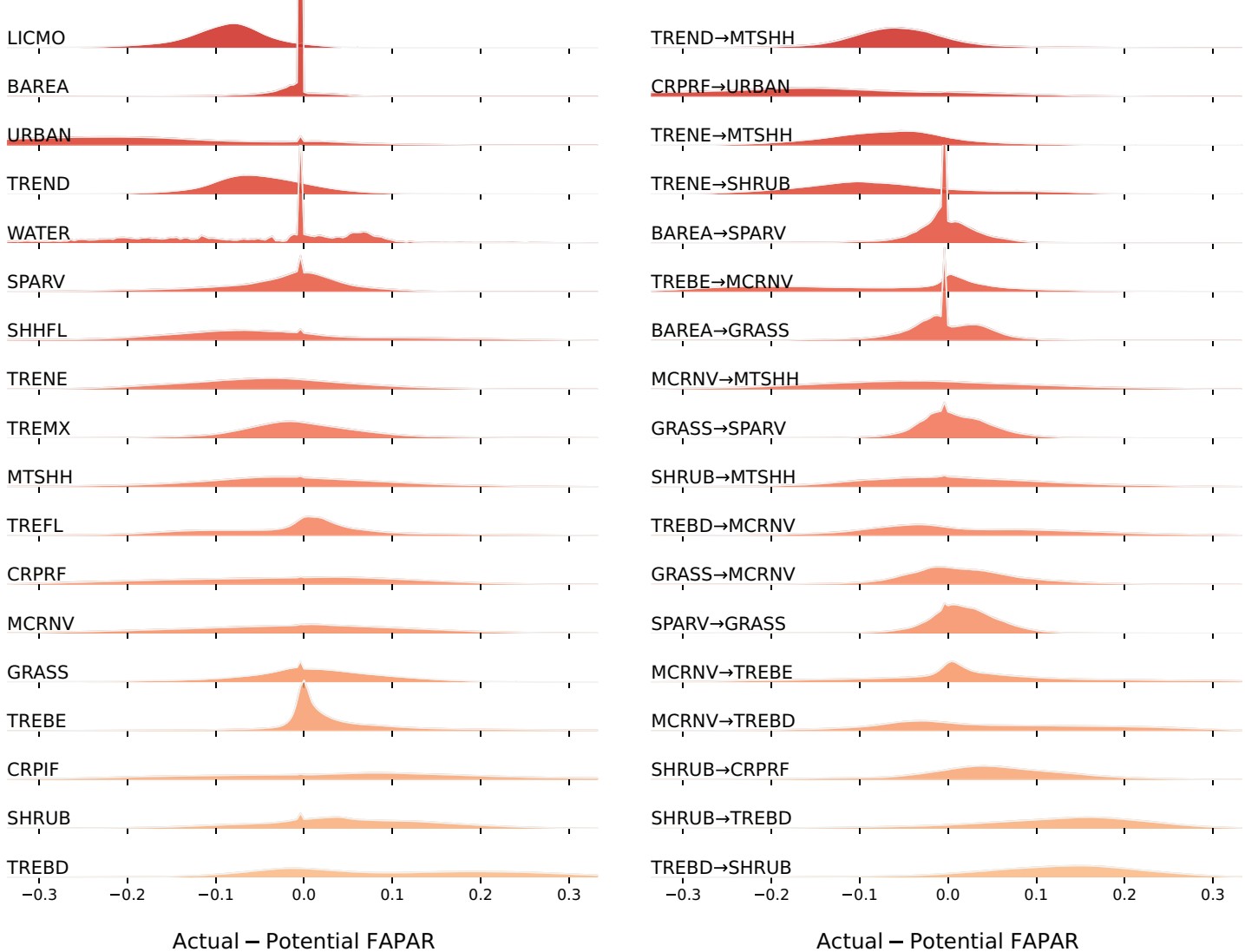

**Figure 11 Distribution of FAPAR gap (ridgeline plots) across stable land cover areas and top biggest land cover change classes.** LICMO—Lichens and mosses; BAREA—Bare areas; URBAN—Urban areas; TREND—Tree cover needleleaved deciduous; WATER—Water bodies; SPARV—Sparse vegetation; SHHFL—Shrub or herbaceous cover flooded; TRENE—Tree cover needleleaved evergreen; TREMX—Tree cover mixed leaf type; MTSHH—Mosaic tree and shrub or herbaceous cover; TREFL—Tree cover flooded; CRPRF—Cropland rainfed; MCRNV—Mosaic cropland or natural vegetation; GRASS—Grassland; TREBE—Tree cover broadleaved evergreen; CRPIF—Cropland, irrigated or post-flooding; SHRUB—Shrubland; TREBD—Tree cover broadleaved deciduous.

## DISCUSSION

### The key variables explaining spatial and temporal variation in FAPAR

The results of modeling actual FAPAR and variable importance assessment in our work clearly show that the key variables that explain the variation in the monthly FAPAR are the length of the growing season, the forest indicator and the annual precipitation. Our results are in line with the findings of *Hengl et al. (2018)* on modelling FAPAR, who applied a similar methodological approach to model FAPAR for the years 2014–2017 based on machine learning and environmental covariate layers. Especially the fact that precipitation

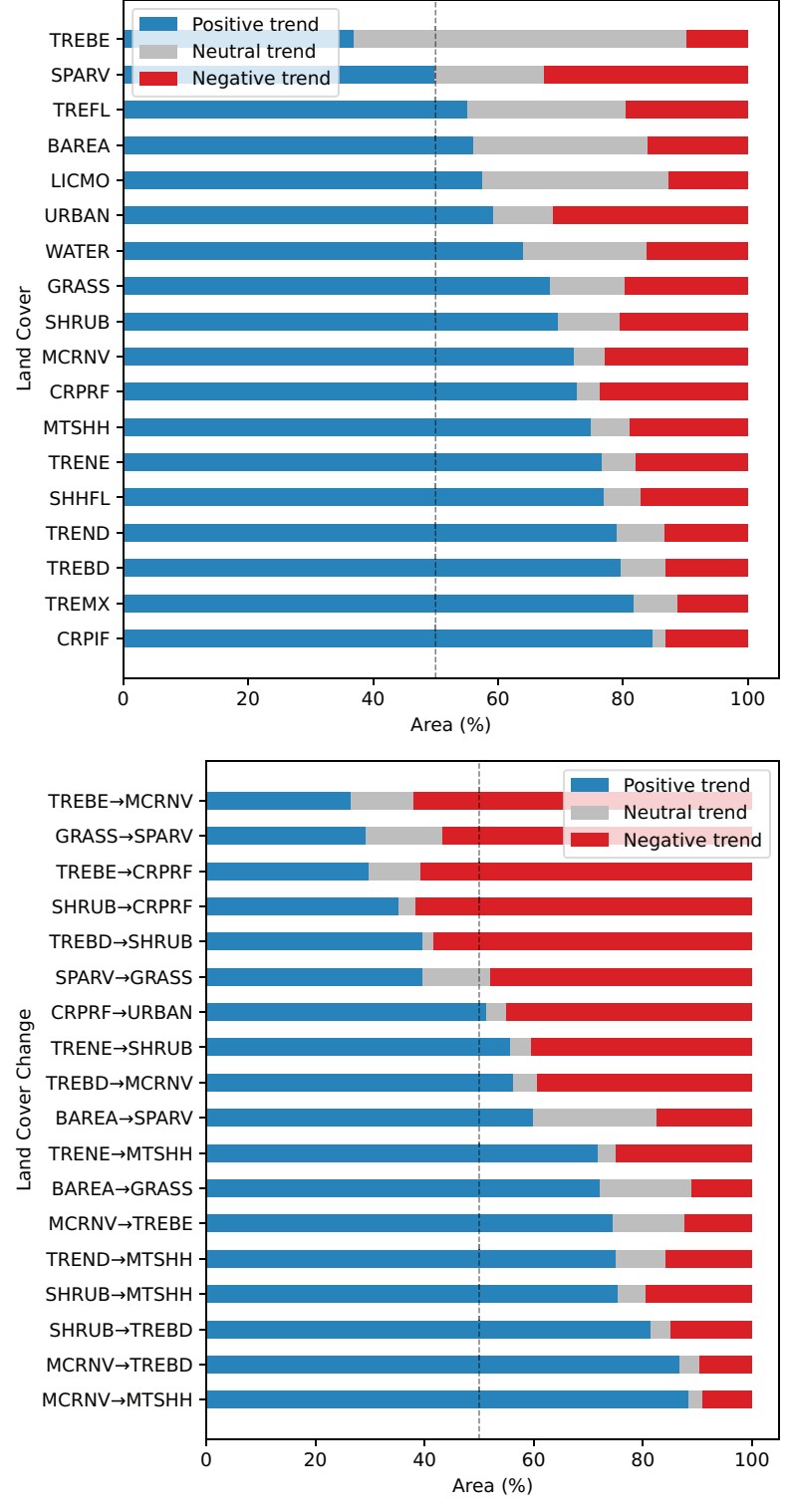

**Figure 12** Summary of FAPAR long–term trend analysis across land cover stable and change areas, shown as percentage within each class. TREBE—Tree cover broadleaved evergreen; SPARV—Sparse vegetation; TREFL—Tree cover flooded; BAREA—Bare areas; LICMO—Lichens and mosses; URBAN— Urban areas; WATER—Water bodies; GRASS—Grassland; SHRUB—Shrubland; MCRNV—Mosaic cropland or natural vegetation; CRPRF—Cropland rainfed; MTSHH—Mosaic tree and shrub or herbaceous

**Figure 12** (continued)
cover; TRENE—Tree cover needleleaved evergreen; SHHFL—Shrub or herbaceous cover flooded; TREND—Tree cover needleleaved deciduous; TREBD—Tree cover broadleaved deciduous; TREMX—Tree cover mixed leaf type; CRPIF—Cropland, irrigated or post-flooding.

is one of the key explanatory variables is confirmed in our study; however, we systematically extended the number of covariate layers and also achieved an overall higher model accuracy.

In our initial data exploration of the monthly aggregated FAPAR and its relationship with EVI, land surface temperature, and water vapor, we found expected relationships between FAPAR and the corresponding variables, such as a decrease in land surface temperature with increasing FAPAR. EVI has been widely used as a measure of vegetation density and an indicator of primary productivity (*Brown et al., 2020*; *Jay et al., 2016*), therefore a close relationship with FAPAR was expected. However, the wide range of present values in all density plots (see Fig. 3) indicates that the estimation of monthly FAPAR from other biophysical MODIS variables is complex. Both monthly FAPAR, EVI and LST seem to be unique biophysical variables and are all important for global modeling efforts.

## Accuracy of the biophysical models explaining FAPAR

Our results further show that FAPAR can be modeled to a maximum $R^2$ of about 0.9 (based on strict cross-validation). This means that there is still a significant part of the variation (about 10%) that we could not map on this scale of work. Our validation of the aggregated monthly FAPAR against ground reference measurements shows agreement with the validation of GLASS FAPAR V6 by *Ma et al. (2022)* against ground measurements, who reported an $R^2$ of 0.8, but for their 8–day product. The validation of our FAPAR model predictions show high agreement with ground reference measurements as well ($R^2 = 0.75$). The close match between the weekly and monthly model accuracies is possibly due to the relatively smooth variation of FAPAR.

Validation of FAPAR values in *Ma et al. (2022)* using ground reference measurements (GBOV) showed an overestimation of FAPAR at low values. This is also in agreement with *Brown et al. (2020)*, who reported an overestimation of FAPAR derived from remote sensing of sparse vegetation. Additionally, the results of the validation of *Ma et al. (2022)* confirm the overestimation of FAPAR at forest sites, which may be related to the saturation of FAPAR in dense canopies.

The ensemble and the individual base models showed comparable model performance in our study. The ensemble approach, however, offers the main advantage of providing the model deviance, *i.e.*, the standard deviation of the base models, per pixel. In addition, using an ensemble model could potentially increase the spatial consistency of uncertainty as some base learners may perform better or worse in specific feature spaces than others, as has been shown for land cover classes by *Witjes et al. (2022)*. While the tested

hyperparameter sets were limited, we achieved high accuracy. Nonetheless, testing a wider range of values could potentially improve results even further.

Visual inspection of the differences between potential and actual FAPARs (around the world) largely matches our expectations. This indicates that our models can be considered stable and could potentially also be used to predict future FAPAR values, *e.g.*, given different climate change scenarios or similar (as in *Bonannella et al. (2023)*). Similarly, in future research, it will be interesting to model and understand the key drivers of the change in FAPAR in a complex Earth system with changing climate, changing policies, and shifting vegetation.

## Mapping gaps between actual *vs.* potential FAPAR

Our results of comparing actual *vs.* potential FAPAR show that the biggest gaps between the two are found for the following land cover classes:

- Potential FAPAR bigger than actual (negative gap): urban, needle-leave deciduous trees, flooded shrub or herbaceous cover, and lichen and mosses.
- Potential FAPAR smaller than actual (positive gap): broad–leaf deciduous trees, shrubland, irrigated or post–flooded cropland and broad-leaf evergreen trees.

In the above section, we only list statistics based on the annual average of 2021, and monthly values might differ. Analysis of monthly values and mapping of gaps is complex. More robust visualization tools and analyzes are needed to further detect gaps and patterns over time in these complex data.

Also note that in this article we used the concept of "Potential Natural Vegetation", which requires some discussion. With PNV concept we often assume that "natural" is the optimal state of the ecosystem. In reality, there will be many situations (especially where people intensively irrigate the land and make intensive fertilizer applications), where a human-managed or human-designed landscape can show much higher productivity than a "natural" one. Similarly, the actual FAPAR greater than its potential value does not necessarily refer to land recovery or improvement in land degradation status (*Bai et al., 2008*; *Cherlet et al., 2018*). We also observed, for example, that savannas often have a lower potential FAPAR than the actual one. While there is controversy about the term of what is considered *"natural"*, we believe that it is still a useful exercise to quantify the indirect effects of removing the impacts of urbanization, intensive fertilization, and irrigation on potential FAPAR.

## Relationship between long-term trends and gaps in FAPAR in relation to land cover changes

In the last 20+ years, researchers have, without doubt, discovered that the planet is becoming greener. Especially China and India are showing a large overall increase in greenness due to extensive agricultural applications (*Chen et al., 2019*). However, *Gao et al. (2023)* have analyzed NDVI time-series data for 1982–2015 and have found notable increases in interannual variability (IAV) of NDVI over the northern high latitudes, Eastern Europe, and Central Australia, indicating that parts of the world are becoming

more sensitive to climatic, environmental and anthropogenic changes. The greening of the Earth in the last two decades is likely driven by $CO_2$ fertilization (*Chen et al., 2022*), N deposition, climate change, and change in land cover (*Zhu et al., 2016*). However, croplands in China and India appear to have a decrease in the IAV of vegetation greenness, suggesting more controlled management practices through inter-cropping, irrigation and fertilization. Likewise, there is an increase in frequency, severity, duration, and scope of global droughts, so it is likely that the greening of the planet will not continue to compensate for the increased IAV and may even reverse in the coming years (*Chen et al., 2022*).

Our results also show that, for most of the planet (67%), there seems to be a positive increase in FAPAR over the time period 2000–2021. The results of matching the FAPAR trends (betas) with land cover change categories show that stable vegetation cover classes with the largest percentage of negative trends are: sparse vegetation, rain–fed cropland, and mosaic croplands. This could be expected, since these land cover types are potentially especially sensitive to changing climate, such as increased droughts. The most negative mean beta found was −0.14. The highest percentages of negative gaps are found in urban areas and in the vegetation classes: lichens and mosses, needle-leaf deciduous trees, and flooded shrub or herbaceous cover.

In summary, the planet over the 2000–2021 time period has on average been getting greener, but there are many local areas, especially in the tropics and areas of savanna and steppe, where there are distinct decreases in FAPAR over the last 20+ years. Attention needs to be paid to these areas to support land owners and governments in their efforts to restore, reverse and reduce degradation. Despite a general trend toward greening, we can see large gaps in the actual FAPAR around the world.

## Limitations of this study

Although the FAPAR product by *Ma et al. (2022)* is to our knowledge the most comprehensive and longest period product at 250 m spatial resolution, the data set also comes with some limitations:

1. The resolution of images is relatively coarse, which means that many land use systems, especially in regions where crop fields are relatively small, are combined/mixed-in, making it difficult to untangle driving factors. Moving to using Landsat as the primary source of the FAPAR assessment could help produce data with a minimum number of "mixels" (pixels of mixed land use) and increase compatibility with the best global land cover products, *e.g.*, *Zhang et al. (2021)*.

2. It is well known that optical remote sensing sensors have limited accuracy in tropics due to clouds and saturation over dense canopies; hence there is possibly a significant difference in accuracy of all products we produced across the globe, with the tropics probably being the most difficult to track. To avoid effects of clouds and water vapor on estimates of FAPAR we used the p0.95 monthly values; however, in some tropical areas are >80% of time under cloud cover and, hence, probably not even the p0.95 monthly value helps.

3. GLASS FAPAR V6 is derived itself from a machine learning model (Bi–LSTM) based on remote sensing data, and hence introduces an inevitable uncertainty that will propagate to the prediction of potential FAPAR. However, comparison with GBOV ground reference measurements showed that our model predicts actual FAPAR with relatively high accuracy, despite the fact that we did not use surface reflectance values/indices.

Verger et al. (2023) have recently produced smoothed and gap-filled time series of global FAPAR at a spatially coarser resolution of 1 km. This data set overlaps a great deal with the work of Ma et al. (2022) and could potentially also be used to perform a similar analysis. However, we focused in our study on finer resolution data in order to attempt to capture small-scale changes, for example, of land use systems with individual land parcels that are relatively small in size and fractured. In future work, modeling potential FAPAR at even finer spatial resolutions of 30 m will be useful to match the recent highest resolution dynamic land cover products (Potapov et al., 2022; Zhang et al., 2021).

Regarding our modeling results, we highlight the following limitations of the methodology and interpretation of the results:

1. Our cross-validation results indicate that there is still a large component of variation (about 10%) in monthly FAPAR that we can not explain. This is probably a limitation of the moderate spatial resolution we worked with, including using covariates at coarser resolution (1 km) than the FAPAR product. In addition, variable importance can be related to the training sample distribution, possibly under-representing areas of some of the land form or lithologic variables, which are less present on the globe.

2. Although the EML model deviance can give an indication of the uncertainty of the prediction, the lack of ground-truth data on potential FAPAR requires the results to be interpreted cautiously. A more robust framework is needed to calculate the probability distributions of FAPAR per pixel (Andriuzzi et al., 2022).

3. For the prediction of potential FAPAR, we make use of a cross-walk table from the ESA CCI land cover towards the BIOME 6000 classification. This may introduce a bias in the prediction. Furthermore, it results in distinct potential FAPAR boundaries between adjacent biomes, which are unlikely to be so distinctly present in reality.

4. We refer here to the potential FAPAR as the FAPAR under no-human pressure by removing the impact of urbanization and intensive agriculture. This refers to the least human pressure under current conditions of the month we predict on. We again highlight the "shifting baseline syndrome" (Papworth et al., 2009), where despite using a global coverage of training points, the results may also be subjected to this syndrome. We cannot preempt that there is human pressure present in areas that are not indicated by our variables, and there may be legacy effect of preceding human pressure (since likely most of the globe has been subjected to human influence at some point).

5. Climate data sets we used are only an approximation of true conditions; climate depends also on present vegetation type and will change if there is a forest where there has been none before, and vice versa; therefore, separation of climate and vegetation is not trivial.

Here we assumed that climate is an independent variable for FAPAR modeling, but this is likely an oversimplification, especially at finer resolutions.

6. FAPAR does not just depend on the current month climate/precipitation, but also on preceding conditions, *e.g.*, the cumulative precipitation of 2–3 months before. We have so far ignored this aspect in this study, as it would increase the computational complexity and would require additional testing.

7. The most important variables in our results were not human pressure variables, which overall have small influence on the prediction, although clearly shown in irrigated areas. The forest cover indicator, on the other hand, had a high importance for the prediction of potential FAPAR.

8. Vegetation cover variables based on ESA CCI land cover change maps are at 300 m resolution. We noticed that GLASS FAPAR V6 picked up a deforestation signal at some sites in Brazil (Fig. 9), but ESA CCI land cover did not show a change, which also affects the model training.

Finally, there are two global vegetation modeling paradigms in the literature: (1) the Dynamic Global Vegetation Models (DGVMs), and (2) data science models based on fitting ML models to training data. DGVMs such as the Lund–Potsdam–Jena-DGVM, have been used to simulate potential primary productivity by excluding the land-use component from the process-based model (*Haberl et al., 2007*). DGVMs are generally restricted by the simplified mechanistic rules set in the model and the definitions of vegetation (*Scheiter, Langan & Higgins, 2013*). Data-driven ML approaches can be used to estimate the possible potential state of vegetation based on correlations with biophysical variables (*Hengl et al., 2018*). In this article, we used a data-driven approach, but this comes at the cost that it is purely correlative, *e.g.*, based on an extrapolation of observed relationships and not on modeling actual processes.

## Implications of this study and next steps

With the FAPAR analysis produced and a global monthly map of the potential FAPAR, we have tried to contribute to international programmes such as Land Degradation Neutrality (UNCCD) and the Sustainable Goals of the United Nations (UN SDGs). We have released our data as open, and our code (used to produce models, visualizations, and predictions) *via* GitHub (https://github.com/Open-Earth-Monitor/Global_FAPAR_250m). We hope that this will inspire other groups to review our results and expand our research in new directions. We especially hope that national and local agencies will find these data as a good reference for spatial planning and that many small medium enterprises will see an opportunity for practical applications and products.

Although this work has taken significant time and computing resources, many sections of modeling and assessment could be further improved. Some potential extensions of this research include:

1. For a more thorough understanding on the impact of modifying the human pressure variables for the potential FAPAR prediction results, a sensitivity analysis of the model

could be performed, *i.e.*, analysing the effect each of them has individually but also in synergy with each other. The interaction between human pressure variables and climate variables could be investigated to better understand the magnitude of uncertainty in potential FAPAR.

2. In future studies, it would be interesting to investigate ways to quantify the "shifting baseline syndrome", and to include these estimates in the modeling of potential FAPAR. As an example approach, we could think of finding ways to model FAPAR for historic times and use these values as a reference (*Enquist et al., 2020*). In addition, it would be interesting to compare our results with an approach not reliant on any hard class distinction of biomes or land cover types to account for the spatial gradient change of vegetation cover. One way could be to include fractions of plant functional types, for example recently made available by ESA CCI (*Harper et al., 2023*).

3. Due to limited computing resources, we had to subset the globe to a set of training points as input for modeling. It would be interesting to increase the size of the training points and compare the results with our current model. In our approach we made use of a linear regressor as meta-learner. The deployment of other meta-learners, such as logistic regressors, could potentially improve results further. In addition, to get a more robust way to estimate uncertainty of the EML predictions, it would be interesting to test the conformal prediction (*Johansson et al., 2014*) for this purpose.

4. For the land cover change analysis, we compared the years 2000 and 2020 to derive change classes. Some areas might experience multiple changes within this time series, which could be further investigated in following studies.

5. To derive robust conclusions on the degree of land degradation, additional information from environmental variables should be included to supplement the data on land productivity, such as land cover change or soil organic carbon. *Shao et al. (2024)*, for example, made use of specific variables, such as wind speed, known to drive land degradation in their local study area. Our analysis of land productivity could be used as one of the variables in such frameworks that make use of the "convergence of evidence" to indicate potential degradation areas (*Gianoli, Weynants & Cherlet, 2023*).

6. Although our results give estimates of the gap between actual and potential FAPAR for current times, this approach could also be applied to future climate simulations. With estimates of future potential FAPAR, we could include this information already today in restoration, mitigation, and adaptation planning.

## CONCLUSION

Land degradation is a global challenge threatening functioning of ecosystems and human livelihoods. To identify and mitigate these challenges, highly-resolved information of the current potential states of vegetation is usually required. The potential vegetation FAPAR at 250 m resolution we produced in this article gives an indication of conditions in the absence of urbanization or intensive agriculture. We applied a data-driven framework using ensemble machine learning to estimate the gap of actual and potential FAPAR. Our results show that FAPAR can be explained with high accuracy ($R^2 = 0.89$) using

environmental and human pressure variables. We demonstrate the application of this framework for the year 2021. The resulting predictions of potential FAPAR not only highlight the gaps, but also show that the model captures irrigation areas, where we expect the actual FAPAR to be higher than the potential FAPAR.

With the produced map of the FAPAR gap, and in combination with estimates of the long–term trend, we identified areas of large negative FAPAR gaps as well as decreasing actual FAPAR values over time and *vice versa*. Trend analysis showed that while large areas of increased FAPAR can be observed, sparsely vegetated areas and vegetation depending on precipitation display largely negative trends. Our assessment specifically showed large negative FAPAR gaps (actual lower than potential) for classes urban, needle-leave deciduous and flooded shrub or herbaceous cover, while strong negative FAPAR trends were found for classes urban, sparse vegetation and rainfed cropland. On the other hand, classes irrigated or post-flooded cropland, tree cover mixed leaf type, and broad-leave deciduous showed largely positive trends. In further research, other variables that support the indication of land degradation in a specific context should be integrated into this framework (such as water table, soil organic carbon or land cover change). This methodology could be applied as a general framework for estimating and visualizing land degradation. Applying this framework to the entire time series of 2000–2021, we could track changes in the FAPAR gap spatially and temporally.

Although our approach requires significant computational resources, it seems to produce stable results for the whole land mask. Its main advantages are: (1) a single model is fitted for the complete space-time cube so that predictions are consistent; (2) potential land degradation is assessed from two aspects, *i.e.*, as a declining trend in FAPAR and as a difference between actual and potential vegetation, allowing land managers to consider multiple land restoration options. In subsequent work, this methodology could also be tested and applied to future simulations of potential FAPAR based on different climate change scenarios, which would allow us to include this information already today in our restoration planning.

## ACKNOWLEDGEMENTS

The authors are grateful to Sytze de Bruin and Kirsten de Beurs (Wageningen University & Research) for their valuable feedback on the methodology and design of this study. We also thank Martijn Witjes and Carmelo Bonannella (OpenGeoHub) for providing suggestions on the methodology.

This study has been undertaken using *"FIPAR, RM–6"* from GBOV "Ground Based Observation for Validation" (https://land.copernicus.eu/global/gbov) founded by European Commission Joint Research Centre FWC932059, part of the Global Component of the European Union's Copernicus Land Monitoring Service. GBOV products are developed and managed by ACRI-ST with the support from University College London, University of Leicester, University of Southampton, University of Valencia and Informus GmbH. We thank all principal investigators of the NEON, FLUXNET, SM, and TERN networks for the measurements collected in the field and used to generate GBOV products.

### Funding

The Open-Earth-Monitor Cyberinfrastructure project has received funding from the European Union's Horizon Europe research and innovation programme under grant agreement No. 101059548. The Global Pasture Watch under the Land Carbon Lab project has received funding from the Bezos Earth Fund. The funders had no role in study design, data collection and analysis, decision to publish, or preparation of the manuscript.

### Grant Disclosures

The following grant information was disclosed by the authors:
European Union's Horizon Europe research and innovation programme: 101059548.
Bezos Earth Fund.

### Competing Interests

Julia Hackländer, Leandro Parente, Yu-Feng Ho, Tomislav Hengl, Rolf Simoes, Davide Consoli, Murat Şahin, Xuemeng Tian & Ichsani Wheeler are employed by OpenGeoHub.

### Author Contributions

- Julia Hackländer conceived and designed the experiments, performed the experiments, analyzed the data, prepared figures and/or tables, authored or reviewed drafts of the article, and approved the final draft.
- Leandro Parente conceived and designed the experiments, performed the experiments, analyzed the data, prepared figures and/or tables, authored or reviewed drafts of the article, and approved the final draft.
- Yu-Feng Ho performed the experiments, analyzed the data, authored or reviewed drafts of the article, and approved the final draft.
- Tomislav Hengl conceived and designed the experiments, analyzed the data, authored or reviewed drafts of the article, and approved the final draft.
- Rolf Simoes performed the experiments, analyzed the data, prepared figures and/or tables, authored or reviewed drafts of the article, and approved the final draft.
- Davide Consoli conceived and designed the experiments, performed the experiments, analyzed the data, authored or reviewed drafts of the article, and approved the final draft.
- Murat Şahin analyzed the data, prepared figures and/or tables, and approved the final draft.
- Xuemeng Tian performed the experiments, analyzed the data, authored or reviewed drafts of the article, and approved the final draft.
- Martin Jung conceived and designed the experiments, authored or reviewed drafts of the article, and approved the final draft.
- Martin Herold conceived and designed the experiments, authored or reviewed drafts of the article, and approved the final draft.
- Gregory Duveiller conceived and designed the experiments, authored or reviewed drafts of the article, and approved the final draft.

- Melanie Weynants analyzed the data, authored or reviewed drafts of the article, and approved the final draft.
- Ichsani Wheeler conceived and designed the experiments, authored or reviewed drafts of the article, and approved the final draft.

## Data Availability

The data and code produced in this work are available at GitHub and Zenodo:

- Leandro Parente, Davide Consoli, Julia Hackländer, Yu-Feng Ho, & Tomislav Hengl. (2023). FAPAR monthly time-series (250 m): Long-term trend (2000-2021) (Version v20230619) [Data set]. Zenodo. https://doi.org/10.5281/zenodo.8399173

- https://GitHub.com/Open-Earth-Monitor/Global_FAPAR_250m.

The environmental covariate data from CHELSA are available at https://doi.org/10.16904/envidat.332.

The DTM MERIT data is available at https://hydro.iis.u-tokyo.ac.jp/~yamadai/MERIT_DEM.

The land cover data is available at https://www.esa-landcover-cci.org.

The lithologic and land form data is available at Zenodo: Tomislav Hengl. (2018). Global landform and lithology class at 250 m based on the USGS global ecosystem map (1.0) [Data set]. Zenodo. https://doi.org/10.5281/zenodo.1464846.

The human Footprint Index data is available at figshare: Mu, Haowei; Li, Xuecao; Wen, Yanan; Huang, Jianxi; Du, Peijun; Su, Wei; et al. (2021). An annual global terrestrial Human Footprint dataset from 2000 to 2018. figshare. Figure. https://doi.org/10.6084/m9.figshare.16571064.v6.

The nightlight data is available at Zenodo: Hengl (2023). Annual time series of global VIIRS nighttime lights for 2000–2021 at 500-m spatial resolution extrapolated using logistic regression (v0.1) [Data set]. Zenodo. https://doi.org/10.5281/zenodo.7750175.

The population count data is available at the Joint Research Centre Data Catalogue: https://doi.org/10.2905/2FF68A52-5B5B-4A22-8F40-C41DA8332CFE.

The cropland intensity data is available at Zenodo: Bowen Cao, Le Yu, Xuecao Li, Min Chen, Xia Li, Pengyu Hao, & Peng Gong. (2021). A 1 km global cropland dataset from 10000 BCE to 2100 CE (Version 2) [Data set]. Zenodo. https://doi.org/10.5281/zenodo.5759237.

The data on Enhanced Vegetation Index, water vapour and land surface temperature is available at:

- https://doi.org/10.5067/MODIS/MOD13Q1.006,
- https://doi.org/10.5067/MODIS/MOD11A2.006,
- Leandro Parente, Rolf Simoes, & Tomislav Hengl. (2023). Monthly aggregated Water Vapor MODIS MCD19A2 (1 km): Long-term data (2000-2022) (Version v20230808) [Data set]. Zenodo. https://doi.org/10.5281/zenodo.8226282.

The GBOV reference measurements are available at https://land.copernicus.eu/global/gbov/dataaccessLP.

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
