# Peer review of "Land potential assessment and trend-analysis using 2000–2021 FAPAR monthly time-series at 250 m spatial resolution"

_PeerJ, doi:10.7717/peerj.16972_

## Round 0.1 · original submission · Minor Revisions

Most of the issues are very minor and I have provided all reviews below and hope they could be helpful for you to revise and improve.

Reviewer 1 ·

Basic reporting

no comment

Experimental design

no comment

Validity of the findings

no comment'

Additional comments

This study proposed a data-driven framework to assess land productivity from GLASS FAPAR and a set of climatic, topographic, geographic, vegetation cover and human pressure factors data.
I found the article intriguing and the methodology sound, making it a suitable fit for this journal. However, I believe some minor revisions are necessary before accepting it.
1. Please elaborate on the background on the “land potential” at the beginning of the abstract.
2. The link on Line 172 cannot be accessed, please update.
3. Please add more information about the GBOV FAPAR reference data used to validate the monthly aggregated GLASS FAPAR. 1) the spatial resolution the reference data represents and 2) how to match the dates between the datasets.

Reviewer 2 ·

Basic reporting

Hackländer et al. present an appealing study on modelling FAPAR, predicting its potential distribution and analysing the gap between potential and actual FAPAR. The description is clear, structure is adequate. My only comment in this respect is about referencing PNV:

It is logical to compare potential FAPAR with potential natural vegetation estimations, however, I find the weight of PNV in the introduction too large. A weight should be placed on introducing FAPAR and it is enough to mention PNV, a concept that is similar in certain points. For the definition of PNV, the original source should be cited, i.e. Tüxen 1956.

Tüxen, R. 1956: Die heutige potentielle natürliche Vegetation als Gegenstand der Vegetationskartierung. Angewandte Pflanzensoziologie (Stolzenau), 13, p. 4–42.

Hengl et al. 2018 was only one of the many applications of the PNV concept introduced by Tüxen. Please note: language should not be a barrier in citing, there is also a recent English explanation/core translation published.

Experimental design

Methods description is mostly adequate and clear. I have two questions, however:
1. Please elaborate in more detail what percentiles repesent in this sentences and after:
“What is First, we aggregated the GLASS FAPAR V6 8-day time series from 2000 to 2021 to monthly values of three percentiles (5th, 50th and 95th).. 5th etc percentile”
2. Please justify in more detail the reason to include geometric temperatures. The current explanation is that they represent the shape of the Earth, however, this explanation still not conveys the significance of including them into the models. It would be good to see a sentence or two of explanation why this variable is necessary.

Validity of the findings

Findings appear valid, data are supplied. I have one comment regarding the Discussion:
“Our results confirm the findings of Hengl et al. (2018)” – this comparison does not appear to be informative as we know that the vegetation component of this potential FAPAR model originates from Hengl et al. 2018. Thus the presented model would have to be particularly deviating not to confirm pattern in the input data.

Reviewer 3 ·

Basic reporting

This article is written professional, covering basic introduction, liiustrated workflow and clear conclusion. Necessary literature references and sufficient background are provided. Results are clear with structured figures, tables, and repo.

Experimental design

The experimental design is reasonable and easy to follow, from the general workflow, introduction of the FAPAR data, and considerable explanations of predictor variables. The four research questions are well defined and meaningful. The provided Meta-learning approach is novel and interesting. Within that, the framework is able to fill the knowledge gap in this field.

Comment: for the baseline models in table 4, the size of both tree models and neural networks are pretty small, i.e. 44 estimators for trees and 4 layers & 128 neurons for NNs. The lift of meta model might be mitigated when the model size gets large.

Validity of the findings

Baseline models are well established, including tree-based models and neural networks. Improvements are observed of the proposed EML based on several metrics. All data and codes have been provided and looks easy to apply. Also, Conclusions are well stated by an interesting gap and trend analysis.

Comment: The ensemble lift based on evaluation metrics in table 4 is not significant. Besides linear models, some other meta-learners could be considered. Also, other metrics like MAPE could be evaluated as well to illustrate the improvement.

Additional comments

Overall, the quality of this manuscrip is high. We should accept it with more details provided.

---

## Round 0.2 · accepted · Accept

Thanks for addressing all of the reviewers' comments, and I think this manuscript is ready for publication. Congratulations!